# Sedimentary pyrite sulfur isotopes track the local dynamics of the Peruvian oxygen minimum zone

Virgil Pasquier [1✉], David A. Fike[2] & Itay Halevy [1]

Sulfur cycling is ubiquitous in sedimentary environments, where it mediates organic carbon remineralization, impacting both local and global redox budgets, and leaving an imprint in pyrite sulfur isotope ratios ($\delta^{34}S_{pyr}$). It is unclear to what extent stratigraphic $\delta^{34}S_{pyr}$ variations reflect local aspects of the depositional environment or microbial activity versus global sulfur-cycle variations. Here, we couple carbon-nitrogen-sulfur concentrations and stable isotopes to identify clear influences on $\delta^{34}S_{pyr}$ of local environmental changes along the Peru margin. Stratigraphically coherent glacial-interglacial $\delta^{34}S_{pyr}$ fluctuations (>30‰) were mediated by Oxygen Minimum Zone intensification/expansion and local enhancement of organic matter deposition. The higher resulting microbial sulfate reduction rates led to more effective drawdown and $^{34}S$-enrichment of residual porewater sulfate and sulfide produced from it, some of which is preserved in pyrite. We identify organic carbon loading as a major influence on $\delta^{34}S_{pyr}$, adding to the growing body of evidence highlighting the local controls on these records.

[1] Earth and Planetary Sciences, Weizmann Institute of Science, Rehovot, Israel. [2] Earth and Planetary Sciences, Washington University in St. Louis, St. Louis, MO, USA. ✉email: virgil.pasquier@weizmann.ac.il

The sulfur cycle is closely coupled to oxygen, carbon, and iron cycles[1,2]. It interacts with the carbon cycle through the process of microbial sulfate reduction (MSR), an anaerobic metabolism in which sulfide forms by reduction of sulfate coupled to the oxidation of organic matter, hydrogen, or methane[3,4]. In porewaters, microbially produced sulfide ($H_2S$ and $HS^-$) can react with ferrous iron or iron oxides to form sedimentary pyrite[1] ($FeS_2$), which may eventually be preserved in the sedimentary rock record. The burial of pyrite represents an indirect net source of oxygen to the ocean-atmosphere and is, therefore, an important factor regulating Earth's surface redox state over geologic time[1,2].

Marine pyrite preserves a sulfur isotope signature ($\delta^{34}S_{pyr}$) of microbial metabolic activity and physical processes of transport and mineralization. Of these processes, the microbially-mediated reduction of sulfate to sulfide carries the largest isotopic fractionation (up to ~70‰)[5–9], whereas other metabolic transformations and mineral precipitation and dissolution typically carry much smaller fractionations[10]. Aqueous sulfide forms primarily by the activity of microbes that mediate organoclastic sulfate reduction (OSR) and anaerobic methane oxidation coupled to sulfate reduction (AOM-SR). Our current knowledge suggests that AOM-SR is most likely performed through a syntrophic relationship between methanotrophic archaea and sulfate reducing-bacteria and may involve various cooperative metabolic strategies[11–13]. The favorable thermodynamics of OSR relative to methanogenesis typically results in vertical separation of these metabolisms; OSR dominates closer to the sediment–water interface (SWI), where the sulfate concentration is relatively high, and methanogenesis takes over deeper in the sediments when sulfate levels decrease[14]. The spatial separation of OSR and methanogenesis naturally leads to partial separation of OSR and AOM-SR, the latter of which occurs mostly at sulfate-methane interfaces. We note, however, that there is great variation in this classical zonation. For example, the concurrent activity of both OSR and methanogenesis, and thus AOM-SR, has been reported in organic-rich marine environments[15,16], including the Peru margin[17]. Both microbial pathways are observed to discriminate against $^{34}S$, with a microbial sulfate–sulfide sulfur isotope fractionation ($\varepsilon_{mic}$) ranging from –3 to –70‰, in both natural and laboratory settings[5–9]. The magnitude of $\varepsilon_{mic}$ is highly dependent on the cell-specific sulfate reduction rate (csSRR), which is set by the availability and nature of electron donors, and the ability of the microbial population to exploit the available substrates[5–9]. Importantly, $^{34}S$-depleted sulfide produced by both OSR and AOM-SR may get incorporated into pyrite and contribute to $\delta^{34}S_{pyr}$ values observed in sediments and sedimentary rocks.

Stratigraphic records of $\delta^{34}S_{pyr}$ are commonly used to reconstruct global sulfur-cycle fluxes and processes (ref. [18] and references therein), though there is increasing evidence for the importance of local processes within the sediment for the preserved isotopic composition of pyrite[18]. A prominent example of local depositional controls on $\delta^{34}S_{pyr}$ records is the large-amplitude stratigraphic variation (>70‰) correlating with ~100 000-year Pleistocene glacial−interglacial cycles in the Gulf of Lion, Western Mediterranean Sea[19], and in an onshore site located on the New Zealand shelf[20]. Given the long residence time of sulfate in the ocean (13 Myr; ref. [21]), such swings in $\delta^{34}S_{pyr}$ cannot reflect changes in the marine sulfate reservoir and must instead reflect inherently local environmental or microbial processes. In the cases in question, glacial intervals were characterized by higher sedimentation rate, and pyrite in these sediments formed in an environment more rapidly isolated from the overlying seawater. In such settings, MSR results in a steep decrease in porewater sulfate concentrations, which is accompanied by a progressive $^{34}S$ enrichment of the residual sulfate

(and product sulfide) with depth in the sediment (i.e., following a Rayleigh-type distillation). Interestingly, over the same time period, a deep basin borehole located only 1000 km away from the onshore New Zealand shelf site preserves no glacial−interglacial $\delta^{34}S_{pyr}$ variation, reflecting deposition in water deep enough that Pleistocene sea-level fluctuations did not cause major sedimentary regime shifts. Instead, a long-term, monotonic $\delta^{34}S_{pyr}$ increase at the basinal site was interpreted to reflect a combination of a decrease in bottom-water temperature and a decrease in reworking intensity and frequency, both driven by changes in deep oceanic currents[20]. Rather than changes in the global sulfur cycle as drivers of variation in $\delta^{34}S_{pyr}$ values, it appears that the Gulf of Lion and New Zealand shelf data highlight the role of local to regional factors (e.g., deposition rate and continuity) in controlling $\delta^{34}S_{pyr}$.

It is clear from the above that local changes in the depositional environment can overprint 'global' signals in $\delta^{34}S_{pyr}$. However, other factors with the potential to influence the isotopic composition of pyrite have not been systematically investigated. Particularly, as sedimentary organic carbon is the main driver of metabolic activity within sediments, including OSR and AOM-SR, its concentration and reactivity are expected to exert a strong local control on the sedimentary pyrite sulfur isotope record. For a given set of environmental and depositional conditions, a higher concentration and/or reactivity of organic carbon supports a larger, more active microbial population and, consequently, higher bulk rates of microbial activity (e.g., sulfate utilization by MSR). To investigate the effects of this potential local driver of pyrite sulfur isotope variation, we focused our investigation on a sedimentary succession from the Peru margin (Fig. 1), where upwelling-driven primary productivity results in the deposition of hemipelagic, laminated sediments rich in organic matter on the shelf and upper slope. The high carbon export to the seafloor allows methanogenesis within the sulfate zone, possibly in organic-rich micro-niches where sulfate is depleted[17,22] (Fig. 1b). As part of ODP leg 201, site 1229 comprises sediment deposited at a water depth of ~150 m, in the core of the modern oxygen minimum zone (OMZ) on the South American continental slope (see Supplementary Note 1 for details on the sedimentary setting). Oceanic biological productivity and OMZ expansion and contraction have been reported to vary in this site synchronously with the 100000-year Pleistocene glacial−interglacial cycles[23,24]. These oceanographic changes led to variations in organic carbon deposition fluxes on the timescale of glacial−interglacial cycles, resulting in strong vertical motion of subseafloor redox zonation and associated microbial communities[22,25–27]. For example, an increase in the load of organic carbon results in a shoaling of all redox boundaries, including the sulfate-methane transition zone (SMTZ). Indeed, the combination of a spike in the concentration of archaeol, a biomarker diagnostic of AOM, with low $\delta^{13}C$ values in porewater dissolved inorganic carbon (–71, –73 and –50‰) ~10 m below the seafloor (hereafter mbsf), far above the depth of the present-day SMTZ (~44 mbsf), have been interpreted as evidence for a shallow and stationary paleo-SMTZ[27] (Fig. 1b).

Here, in contrast to prior work in well-oxygenated marine shelf environments with a relatively steady deposition flux of organic matter[19,20], we investigate the effect of temporally varying organic matter deposition rates and redox conditions on the $\delta^{34}S_{pyr}$ record. We use carbon−nitrogen−sulfur concentrations and stable isotopes of glacial−interglacial sediments to identify strong local environmental controls on sedimentary $\delta^{34}S_{pyr}$. Varying rates of microbial metabolic activity, regulated by the effects of regional variations in the OMZ extent and dynamics on the flux of sinking organic matter, appear to drive the observed $\delta^{34}S_{pyr}$ variability on the Peruvian margin.

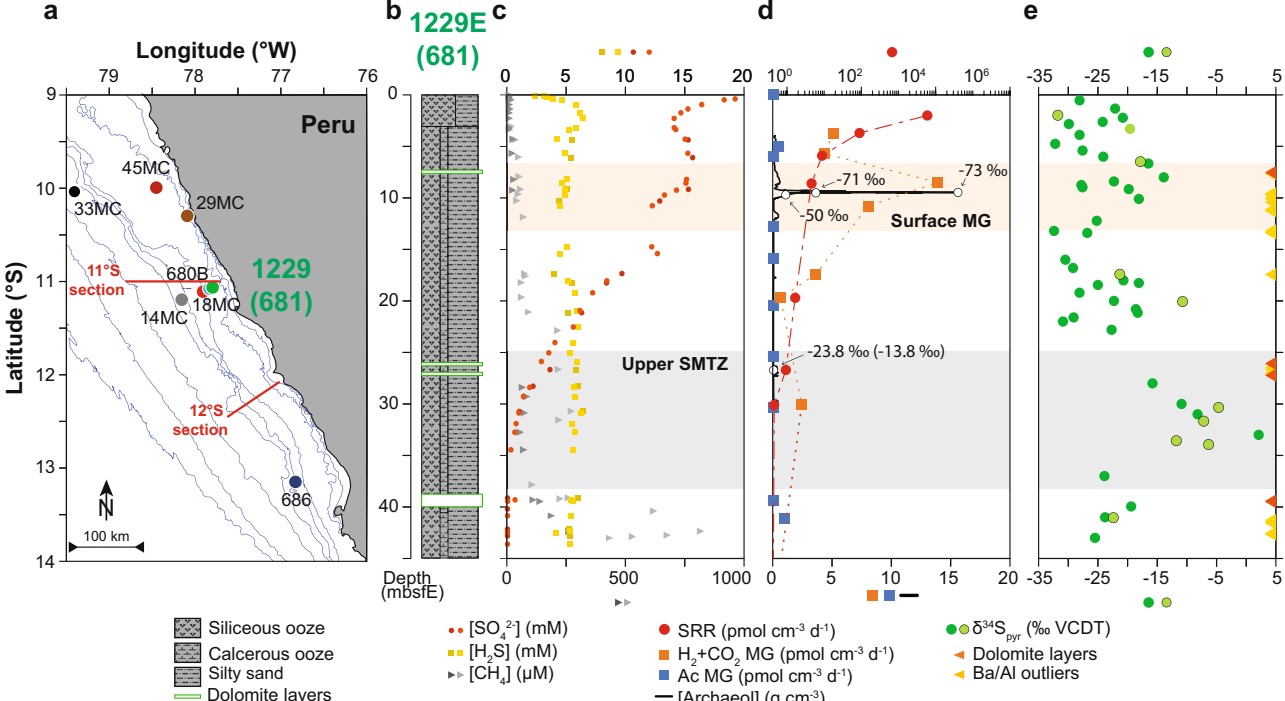

**Fig. 1 Geography and stratigraphy of the study site. a** Geographic location map of samples investigated in this study. Location of the Ocean Drilling Program (ODP) site 1229 (green dot) and other cores mentioned in this study. **b** Stratigraphic section of hole 1229E. Redrawn from ref. [46]. **c** Depth profiles of porewater sulfate (mM), sulfide (mM), and methane (μM) concentrations. Dark and light symbols refer to site 1229 hole A and D, respectively. **d** Sulfate reduction rate (SRR, in pmol cm$^{-3}$ d$^{-1}$), hydrogenotrophic methanogenesis rate (H$_2$ + CO$_2$ MG, in pmol cm$^{-3}$ d$^{-1}$), and acetoclastic methanogenesis rate (Ac MG, in pmol cm$^{-3}$ d$^{-1}$; from ref. [25]), archaeol concentration (g cm$^{-3}$; from ref. [26,27]). **e** δ$^{34}$S values preserved in sedimentary pyrite, as measured in this study (dark green dots) and in ref. [28] (light green dots), and location of dolomite layers (from ref. [35]) and barite layers (based on Ba/Al ratios). Both near-surface methanogenesis (surface MG; orange band) and the sulfate−methane transition zone (SMTZ, gray band) are indicated.

## Results

**Chronological framework.** In order to examine the processes occurring at glacial−interglacial timescales, it was necessary to develop an age model and sampling strategy at a much higher resolution than the timescale resolvable in previous studies[28]. The chronology applied in this study was developed by aligning log(Ca/Ti) data[27] (influenced by carbonate marine productivity and detrital inputs), measured by X-ray fluorescence, with the planktonic foraminiferal δ$^{18}$O record from the Eastern Pacific[29] (TR163-19) for the last 360 ky and with the global benthic δ$^{18}$O stack[30] (Supplementary Fig. 1). Such an approach has previously produced accurate long-term chronologies from multiple locations in which minima in log(Ca/Ti) were shown to correspond to cold stadial intervals in foraminiferal δ$^{18}$O records[31]. We apply this methodology to ODP core 1229E because of the identification of repeated cycles in log(Ca/Ti) and its striking resemblance to the planktonic δ$^{18}$O record from the Eastern Pacific Ocean. The alignment of log(Ca/Ti) measured in ODP core 1229E with the δ$^{18}$O record reveals that the upper 45 m of the sedimentary archive record the last ~610 ky, in agreement with an age model developed for a borehole[32] located several meters from core 1229E (Supplementary Fig. 2). Based on our chronology, sedimentation rates over the majority of the time covered by core 1229E are low relative to the Holocene sedimentation rates derived from $^{14}$C-dating of the upper 4 m of sediment[33]. Importantly, we find only minor glacial−interglacial variations in sedimentation rate (Supplementary Fig. 1).

Comparison with previously published data[25–28,34,35] required correlation between hole 1229E (this study) and holes 1229A and 1229D, which was achieved using core images and magnetic susceptibility measurements (Supplementary Fig. 3). Consequently, all data available at site ODP 1229 can now be placed on the chronological framework developed for hole 1229E and plotted against depth in hole E (mbsfE).

**Sulfur, carbon, and nitrogen isotopes.** We performed 39 pyrite sulfur isotope analyses on the upper 44 m of the ODP 1229E borehole (Figs. 1c and 2; Supplementary Fig. 4). Throughout the core, δ$^{34}$S$_{pyr}$ values show significant variation, from −32.4 to +2.1‰, whereas total reduced sulfur (TRS) contents vary between 0.4 and 1.9 wt.%. No clear relationship is observed between TRS contents and δ$^{34}$S$_{pyr}$ values (Supplementary Fig. 5). Complementary analysis of organic carbon isotopes (δ$^{13}$C$_{org}$) shows variation between −20.7 and −23.4‰, with a negative correlation between δ$^{13}$C$_{org}$ and total organic carbon (TOC) content, the latter of which varies between 0.9 and 6.5 wt.% (Supplementary Figs. 4 and 5). Similarly, organic nitrogen δ$^{15}$N values fluctuate between 3.4 and 9.7‰, and total nitrogen (TN) content varies between 0.03 and 0.8 wt.% (Supplementary Figs. 4 and 5). No relationship is observed between δ$^{15}$N and TN (Supplementary Fig. 5), though there is a scattered positive covariation between δ$^{15}$N and TOC (Supplementary Fig. 5).

**Iron speciation.** The Fe$_{pyr}$/Fe$_{HR}$ ratio in the uppermost sediment sample in this study (at 0.53 mbsf) is 0.53 and increases to values of 0.77 ± 0.05 between 1.33 and 33 mbsf (Supplementary Fig. 4). The Fe$_{HR}$/Fe$_T$ ratio then slightly increases with depth, with a slope of 0.0025 per meter ($R^2$ = 0.17; Supplementary Fig. 4). All samples plot in the uppermost part of the ferruginous or in the borderline euxinic field of ref. [36] (Fig. 3a).

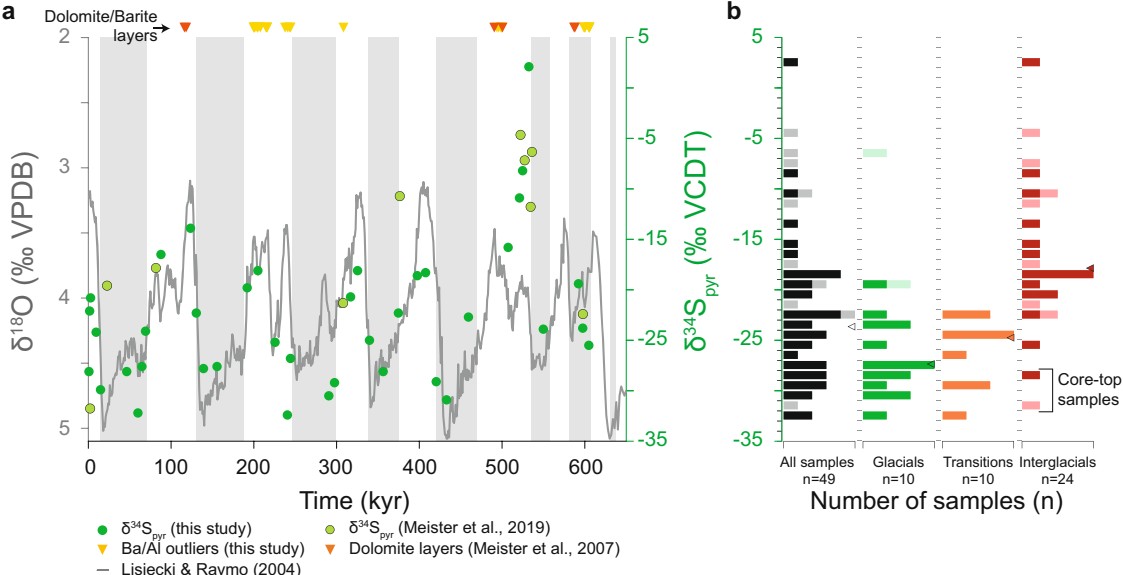

**Fig. 2 The temporal record and distributions of δ34Spyr at ODP 201—site 1229. a** δ34S values preserved in sedimentary pyrite over the last 610 ky, as measured in this study and in ref. [28]. For the global climatic context, deep-sea δ18O values from benthic foraminifera[30] are shown in gray. **b** Histograms of δ34Spyr preserved in Peruvian margin sediment under different climatic conditions. Histogram means are shown by triangles. Dark bars refer to this study, whereas light bars correspond to data from ref. [28].

## Discussion

Within the last 610 ky, δ34Spyr exhibits an apparent bimodal distribution with respect to glacial−interglacial cycles (Fig. 2). Minima occur during glacial periods and are characterized by relatively low isotopic variability (average δ34Spyr = −28.9 ± 1.8‰; n = 14; Fig. 2b). Interglacial samples record maxima in δ34Spyr and larger isotopic variability (average δ34Spyr = −19.7 ± 4.1‰; n = 18; Fig. 2b). A clear linear correlation between δ34Spyr values and foraminiferal δ18O values (Supplementary Fig. 6; $R^2$ = 0.61) prompts us to suggest that δ34Spyr values and climate covary in the studied core. As the timescale of these stratigraphic fluctuations is much shorter than the residence time of seawater sulfate (13 million years), such variations necessarily reflect control by local to regional aspects of the diagenetic environment and/or microbial activity.

Large variations in δ34Spyr values have been observed and attributed to aspects of the local depositional environment, such as bioturbation[37,38] and sediment remobilization[39,40]. Of particular relevance to the present study, previous glacial−interglacial swings of up to 70‰ in δ34Spyr values have been attributed to regional changes in the sedimentation rate[18–20,41,42], also a parameter of the depositional environment rather than the global sulfur cycle. Specifically, strong enrichments in 34S were interpreted to result from higher glacial sedimentation rates, leading to rapid burial relative to diffusive exchange between porewater and seawater, and rapid drawdown with a depth of porewater sulfate by MSR. While the depositional environment of the Peru margin was also affected by sea-level variations (mostly in terms of siliciclastic vs. carbonate sedimentation), there is no strong glacial−interglacial variation in sedimentation rate nor a relationship between sedimentation rate and δ34Spyr (Supplementary Fig. 5j). Moreover, despite an early diagenetic origin of the pyrite in both locations, the changes in δ34Spyr observed here (elevated and variable δ34Spyr during interglacial periods) are opposite in phase to those observed previously (elevated and variable δ34Spyr during glacial periods, associated with higher sedimentation rates). It seems that environmental mechanisms other than those associated with changes in sedimentation rate locally (or regionally) control the sedimentary pyrite record on the Peru margin.

Possible alternative drivers include changes in the redox state of the SWI or changes in the organic carbon loading of the sediments. Both alternatives are expected to affect the relative rates of metabolic activity of sulfate reducers (OSR and/or AOM-SR) as well as the processes of pyrite formation. Next, we explore our geochemical and isotopic measurements together with those made in previous studies to identify the drivers of sulfur isotopic variability in the Peru margin sediments and, likely, other organic-rich marine sediments.

Over the Peru margin and in other sedimentary environments, the reaction of sulfide produced by MSR with various particulate and dissolved iron species to form pyrite is recorded by iron speciation data[32,33] (Fig. 3a). The ratio of pyrite iron to total highly reactive iron, Fepyr/FeHR (also termed the "degree of pyritization"[43]), tracks the extent to which particulate iron species that are reactive towards sulfide (mostly iron oxyhydroxides) are transformed to pyrite by various pathways, all of which require exposure to dissolved sulfide. The available iron speciation data from intra-OMZ sites suggest that pyrite formation is very early[28,44]. Even samples from the SWI display Fepyr/FeHR ratios as high as ~0.25 (Fig. 3a), though no pyrite formation is documented in the water column[45]. Below-detection aqueous sulfide concentrations at the SWI[46] likewise suggest pyrite formation in the upper few centimeters of sediment rather than in the water column, though a contribution of episodic water-column sulfidic conditions[45,47,48] to early pyrite formation cannot be ruled out. Core-top samples (upper 0.5 m of sediment cores) span a range between the Fepyr/FeHR at the SWI (~0.25) and ratios as high as ~0.85 (Fig. 3a). A plateau of 0.8 ± 0.1 in Fepyr/FeHR measured in OMZ settings is usually reached by about 0.3 mbsf, indicating the rapid transformation of reactive iron to pyrite in all cores under the influence of the OMZ. We note that despite Fepyr/FeHR ratios above the traditional threshold for classification of an environment as euxinic (0.8), the water column at the study site is anoxic, but not euxinic.

Iron speciation measurements deeper within the sediments from the Peru margin are only available in borehole 1229 (ref. [28] and this study). Here, Fepyr/FeHR values stay essentially constant between depths of ~0.5 mbsf and the base of the borehole at ~190

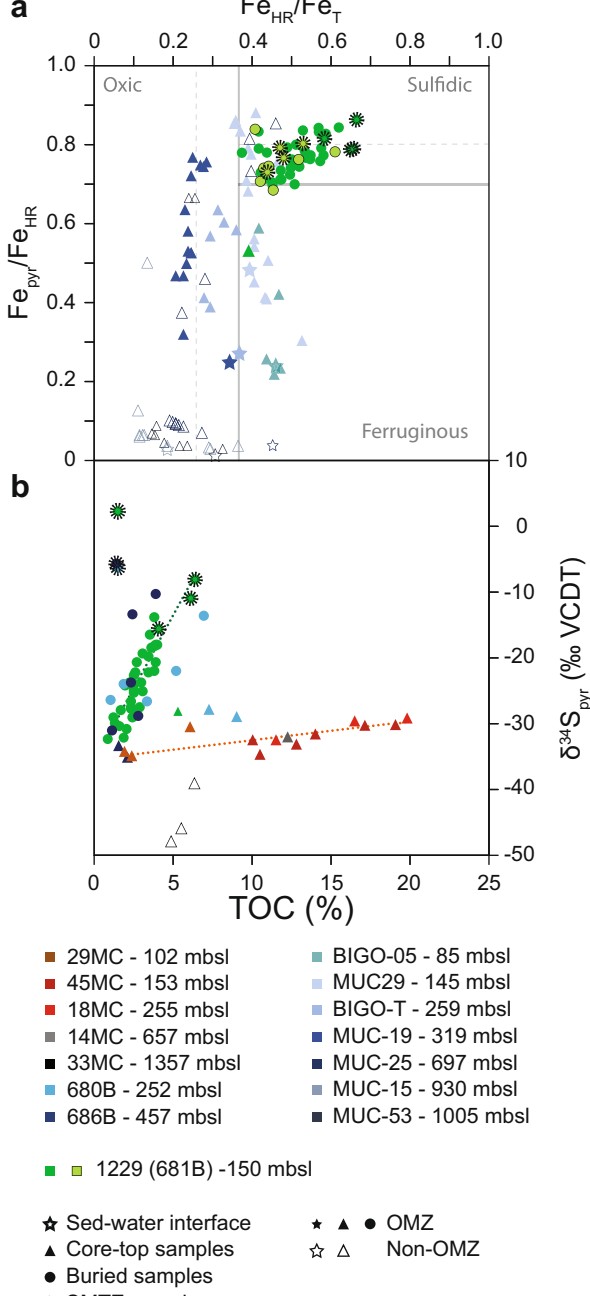

**Fig. 3 Fe speciation-TOC-$\delta^{34}S_{pyr}$ systematics. a** Cross plot of the ratio of pyrite Fe to highly reactive Fe ($Fe_{pyr}/Fe_{HR}$) against highly reactive Fe to total Fe ($Fe_{HR}/Fe_T$) in sediment–water interface (0–1 cmbsf; stars), core-top[28,44] (<50 cmbsf; triangles) and buried[28,44] (>50 cmbsf; circles) samples. Horizontal and vertical lines indicate the proposed boundaries for distinguishing euxinic from ferruginous and oxic water columns[43,69], where solid lines are at $Fe_{pyr}/Fe_{HR}$ and $Fe_{HR}/Fe_T$ recommended on the basis of modern environments, and dashed lines are suggested boundaries for ancient sediments. **b** $\delta^{34}S_{pyr}$ vs. TOC for core-top[51] and buried (this study) samples. Dotted lines represent the trendlines observed between Intra-OMZ core-top and SWI $\delta^{34}S_{pyr}$ values and TOC (orange) and down-core $\delta^{34}S_{pyr}$ values and TOC (green). Samples located within the upper SMTZ are outlined in dotted black. Filled symbols represent cores located within the OMZ, whereas open symbols represent cores outside the OMZ.

mbsf (Supplementary Fig. 4), with an average value of 0.77 ± 0.06. The similarity of this value to that observed in core-top samples from other intra-OMZ locations and its near-constancy throughout the core leads us to conclude that iron-sulfide minerals, which are ultimately preserved as pyrite in the geologic record, mostly form over the first tens of centimeters below the SWI. Iron speciation data and sulfur isotopic compositions of acid-volatile sulfur (a proxy for iron monosulfides), methanol-toluene extractable sulfur (a proxy for zero-valent sulfur), and pyrite similarly suggest early pyrite formation in the relatively organic-rich sediments in the Bornholm Basin[49]. We note that SWI and core-top samples from extra-OMZ sites (in this case, below the OMZ) show much lower $Fe_{pyr}/Fe_{HR}$ values than intra-OMZ sites, usually <0.1 (Fig. 3a), and generally show lower $Fe_{HR}/Fe_T$ values.

There is a striking difference between SWI and core-top $\delta^{34}S_{pyr}$ values measured in sites within the Peru OMZ and those outside the OMZ. The $\delta^{34}S_{pyr}$ values in intra-OMZ sites range from approximately −35 to −30‰, positively correlating with TOC concentrations (Fig. 3b). In contrast, available extra-OMZ $\delta^{34}S_{pyr}$ values are as low as about −50‰. We suggest that this may be an outcome of sulfide accumulation in intra-OMZ sediment pore-waters and an absence of such accumulation in extra-OMZ porewaters. Off the Peru coast, upwelling of oxygen-depleted and nutrient-rich water leads to a near-complete consumption of oxygen in the water column overlying the upper slope and shelf (Supplementary Fig. 7). The resulting OMZ extends from water depths of less than 80 m to roughly 700 m[50]. Where the OMZ intersects the seafloor, TOC concentrations in sediments are higher than those typical of pelagic shelves in non-upwelling regions (Supplementary Fig. 8). Measured $\delta^{13}C_{org}$ values and C/N ratios of core-top sediment are characteristic of fresh to moderately altered marine organic matter[17]. The high concentration of reactive organic matter in intra-OMZ sites exhausts the little available dissolved oxygen relatively close to the SWI. This results in the onset of OSR within the shallow sediments, where sulfate may readily diffuse in from overlying seawater[17,22,25] (Supplementary Fig. 8). In these cases, local production of sulfide at a given depth in the sediment may exceed the rate at which it can be removed by reaction with iron-bearing minerals, and aqueous sulfide will accumulate in the porewater and diffuse away from the depth of its formation (Supplementary Fig. 8). The $\delta^{34}S$ values of sulfide at any given depth will then reflect the mixing of in-situ production and upward diffusion of sulfide from deeper within the sediments. The deeply sourced sulfide forms by reduction of residual sulfate that has experienced a greater degree of isotopic distillation (i.e., more 34S-enriched). Thus, the mixed-source (in-situ and deeper-sourced) sulfide $\delta^{34}S$ values are in all cases higher than those of the instantaneously (locally) produced sulfide. In extra-OMZ sites, both below and above the OMZ, bottom-water O2 concentrations are higher, and the onset of OSR occurs deeper within the sediments, where the amount and reactivity of organic matter are lower. This results in lower rates of OSR, and the kinetics of sulfide consumption (e.g., by reaction with iron) match the rate of sulfide production at a lower porewater sulfide concentration. In this case, the sulfide does not accumulate in porewater, diffusive communication of sulfide between different depths in the sediment is minor, and the isotopic composition of the local porewater sulfide is closer to that of instantaneously generated sulfide. That is, in extra-OMZ sites on the Peru margin, we expect lower porewater sulfide concentrations and lower $\delta^{34}S$ values than in intra-OMZ sites.

Irrespective of its isotopic composition, some of the aqueous sulfide diffuses upwards towards the chemocline, where it is

oxidized to elemental sulfur and sulfate, as shown by the presence of elemental sulfur ($S_{8s}$) and mats of sulfur-oxidizing bacteria (*Thioploca sp.*; refs. [48,51]) near the sediment surface. Some of the porewater sulfide is incorporated into organic matter (by organic matter sulfurization) and/or reacts with iron-bearing minerals to form iron monosulfides (FeS) and pyrite[28,44,51]. The sulfur iso-topic composition of all of the solid products of sulfide reaction ($S_8$, FeS, organic sulfur, and pyrite) is expected to be related to the isotopic composition of the porewater sulfide. Indeed, within the OMZ and over the depth of pyritization, elemental sulfur, FeS, and pyrite show similar $\delta^{34}S$ values of $-31.6 \pm 2.3‰$ (ref. [51]), confirming that their isotopic composition is inherited from a single reservoir of sulfur. We suggest that in intra-OMZ sites on the Peru margin the isotopic composition of this reservoir reflects pooling and upward diffusion of the sulfide product of sulfate reduction (i.e., with higher $\delta^{34}S_{pyr}$ values, near $-30‰$), whereas in extra-OMZ locations the pyrite (and reduced sulfur compounds) form from a reservoir that is closer in isotopic composition to the instantaneously generated sulfide (i.e., with lower $\delta^{34}S_{pyr}$ values, near $-50‰$; Fig. 3b).

Intra-OMZ core-top and SWI $\delta^{34}S_{pyr}$ values positively corre-late with the concentration of TOC (Fig. 3b orange dotted line; slope = 0.25 ‰/% TOC, $R^2 = 0.64$). We suggest that with higher TOC concentration, maximal rates of sulfate drawdown (and isotopic distillation) by OSR are higher, favoring accumulation of the sulfide product and sulfide $\delta^{34}S$ values closer to those of the sulfate source. The relatively shallow slope suggests that the availability of organic substrates in this environment affects the overall rate of OSR, but that other factors may set stronger lim-itations to the activity of sulfate reducers.

Down-core $\delta^{34}S_{pyr}$ values evolve from near-core-top values (~ $-30‰$) to values as high as ~ $-10‰$ for non-SMTZ samples and ~$+2‰$ for SMTZ-related samples (the effect of SMTZ formation and dynamics on pyrite sulfur isotopes is discussed below). The down-core $\delta^{34}S_{pyr}$ values correlate with TOC concentration, forming an array with a much steeper slope (Fig. 3b green dotted line, slope 4.5‰/% TOC, $R^2 = 0.82$) than the $\delta^{34}S_{pyr}$-TOC array defined by the core-top and SWI samples. The different slope suggests that processes other than the degree of sulfide accumu-lation in porewater contribute to controlling $\delta^{34}S_{pyr}$ values, and we propose that an important process is the TOC-dependent rate at which sulfate is drawn down and isotopically distilled upon gradual burial and isolation from the overlying water column. When TOC concentrations are low (e.g., during glacial intervals at the study site), sulfate drawdown with depth in the sediment is relatively gradual, the porewater sulfate is better buffered in concentration and isotopic composition to seawater sulfate, and $\delta^{34}S_{pyr}$ values are relatively low. In contrast, during interglacial intervals when TOC concentrations are high, rapid sulfate drawdown results in Rayleigh-type distillation and an increase in porewater sulfate $\delta^{34}S$ values, which are less effectively buffered to seawater $\delta^{34}S$ values by diffusive exchange with the water column. The $\delta^{34}S$ values of porewater sulfide (ultimately preserved in pyrite) produced by microbial reduction of porewater sulfate are then also higher. We note that in boreholes 680B and 686B, two sites within the OMZ but at water depths of 252 and 457 m, respectively (compared to a water depth of 150 m at our study site) the down-core $\delta^{34}S_{pyr}$-TOC relationship is similar to our study site (Fig. 3b), suggesting that these arrays and the processes governing them may be common among Peruvian OMZ sites, and perhaps other organic-rich sedimentary environments.

The dependence of SWI and core-top $\delta^{34}S_{pyr}$ values on TOC concentrations and on bottom-water $O_2$ concentrations (i.e., intra-OMZ vs. extra-OMZ) may alternatively be explained by the effect of higher TOC concentrations on csSRR. Higher TOC concentrations may translate into higher rates of organic matter

degradation and associated OSR rates. Lower bottom-water $O_2$ concentrations translate into a shallower onset of OSR within the sediments, and therefore, a higher amount and reactivity of organic matter available for sulfate reducers. Thus, both higher TOC concentrations and lower bottom-water $O_2$ concentrations could lead to higher csSRR. At higher csSRR, the microbial fractionation of sulfur isotopes is smaller[5–9], and the relatively $^{34}S$-enriched intra-OMZ samples may simply reflect higher csSRR in association with higher TOC concentrations. This scenario would imply that the population size of sulfate-reducing microbes increases by a smaller factor than a given increase in TOC con-centration. Such a situation could arise if something other than TOC availability limited the population size (e.g., nutrients). If TOC concentrations and sulfate-reducer population size increased by the same factor, then bulk rates of OSR would increase with increasing TOC concentration, but cell-specific rates would remain invariant.

Estimates of csSRR in the upper 50 cm of the sediments may be obtained from combined estimates of bulk SRR and sulfate-reducer cell density. Combining bulk SRR from the study site[17,22] and cell counts available from sites with similar TOC contents[52–54], estimated csSRR is between 0.1 and 1.0 fmol cell$^{-1}$ d$^{-1}$. For some strains of sulfate reducers[7] (e.g., *Desulfovibrio vulgaris* Hildenorough) the higher csSRR implies large, near-equilibrium $\varepsilon_{mic}$ of ~$-70‰$. For other strains[5,6] (e.g., *Desulfovi-brio DMSS-1*), this csSRR implies $\varepsilon_{mic}$ as small as $-50‰$. The isotopic offset between the $\delta^{34}S$ values of seawater sulfate ($+21‰$) and the reduced sulfur solids (elemental sulfur, FeS, pyrite) would imply $\varepsilon_{mic}$ of ~$-50‰$, a smaller negative number than the sulfide−sulfate equilibrium sulfur isotopic fractionation of ~$-70‰$ at the temperature of the study site. However, two observations raise doubt regarding this explanation of smaller $\varepsilon_{mic}$ due to high TOC-dependent csSRR. First, in the study site and in several other intra-OMZ sites, porewater sulfide concentrations are in the millimolar range (Supplementary Fig. 8), suggesting that the isotopic composition recorded in elemental sulfur, FeS, organic sulfur, and pyrite represents substantial pooling of the sulfide product of sulfate reduction. If this is the case, then $\varepsilon_{mic}$ must be more negative than $-50‰$, the observed isotopic offset of the reduced sulfur solids from seawater sulfate. Second, in-situ sulfur isotope measurements of pyrite formed during the Cre-taceous Ocean Anoxic Event 2 (OAE2) record minimal $\delta^{34}S_{pyr}$ values that are offset from coeval seawater sulfate by 55–57‰ (ref. [55]). The OAE2 pyrite formed in a depositional environment with TOC concentrations as high as ~19 wt.%, similar to some intra-OMZ sites, but higher than the maximal TOC concentra-tions encountered at our study site, where TOC ≤ 6.5 wt.%. Sul-fide pooling in porewater is expected under such conditions, meaning that $\varepsilon_{mic}$ was likely more negative than $-57‰$. Unless the reactivity of the marine organic matter was much lower during OAE2 than in our study site, expected sulfate reduction rates would be higher than in our site (due to higher TOC con-centrations). One might expect, therefore, $\varepsilon_{mic}$ in our site to be even more negative than in the OAE2 study, which is to say, close to the equilibrium fractionation of sulfur isotopes between sulfate and sulfide. In-situ pyrite sulfur isotope measurements may dis-tinguish between the hypotheses of varying $\varepsilon_{mic}$ and different degrees of sulfide accumulation as explanations for the variations in core-top $\delta^{34}S_{pyr}$ values observed in our study site, but such measurements are beyond the scope of the current study.

In addition to the positive correlation between $\delta^{34}S_{pyr}$ values and TOC concentrations ($R^2 = 0.74$), which we propose arises from the influence of TOC concentration and reactivity on community-level MSR rates and the degrees of accumulation of porewater sulfide and isotopic distillation of porewater sulfate, a notable feature of the Peru margin data is a positive correlation

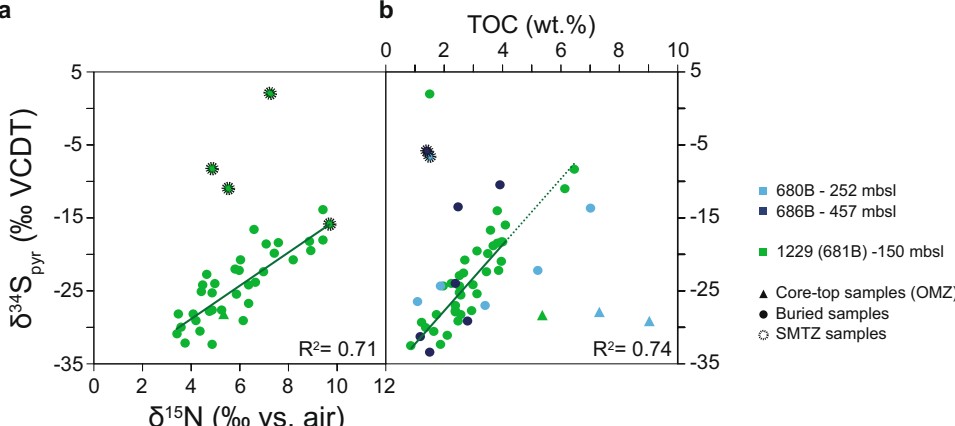

**Fig. 4 $\delta^{15}N$-TOC-$\delta^{34}S_{pyr}$ systematics. a** $\delta^{34}S_{pyr}$ against $\delta^{15}N$. **b** $\delta^{34}S_{pyr}$ against TOC. The three samples located within the modern sulfate−methane transition zone (SMTZ; dotted black outline) are excluded from the linear regressions of $\delta^{34}S_{pyr}$ on $\delta^{15}N$. Marker colors and types are as in Fig. 3.

between $\delta^{34}S_{pyr}$ and $\delta^{15}N$ values ($R^2 = 0.71$; Fig. 4). Such $\delta^{34}S_{pyr}$-$\delta^{15}N$-TOC systematics may provide insight into the local control on $\delta^{34}S_{pyr}$ variations in the Peru sediments. The C−N and $\delta^{15}N$ data are consistent with previously published values and appear to track glacial−interglacial variations in the intensity and/or extent of water-column denitrification over the Salaverry shelf for the last 610 ky[24,46] (Supplementary Fig. 4). Elevated $\delta^{15}N$ values in interglacial sediments are interpreted to reflect more wide-spread suboxic conditions at these times because increased water-column denitrification under oxygen-poor conditions enriches the residual nitrate in $^{15}N$ by up to 20‰ (ref. [56]). The resulting $^{15}N$-enriched nitrate is upwelled, assimilated by phytoplankton, and ultimately preserved in sedimentary organic matter[57].

The more extensive interglacial OMZ, as indicated by our $\delta^{15}N$ data, apparently led to an enhanced flux of reactive organic carbon to the SWI, consistent with the elevated TOC concentrations observed in interglacial strata at the study site. More abundant and more labile organic matter is then available for sulfate reducers, increasing the relative population size and the bulk OSR rate during interglacial intervals. The high availability of reactive organic matter may have also led to the onset of methanogenesis relatively close to the SWI, increasing the interglacial population size and bulk rate of AOM-SR. Indeed, we note that at the study site, hydrogenotrophic methanogenesis was detected at depths within the sediment in which MSR rates were high[17] (Fig. 1b). As little methane (<50 µM) accumulates in surface sediment (Supplementary Fig. 8), the high MSR rates observed in core-top likely reflect a combination of OSR and AOM-SR. As discussed above, higher bulk rates of MSR, irrespective of the electron donor (organic matter or methane), result in more effective drawdown and isotopic distillation of porewater sulfate, and in more pronounced accumulation of porewater sulfide, both of which lead to preservation of higher interglacial $\delta^{34}S_{pyr}$ values.

Heterogeneity in the degree of TOC enrichment and its associated effect on the overall MSR rate and consequent isotopic distillation of porewater sulfate and accumulation of porewater sulfide explains the greater interglacial variability in $\delta^{34}S_{pyr}$ values. The lower and relatively invariant $\delta^{34}S_{pyr}$ during glacial times can be understood as the result of the basinward migration of the OMZ, as attested by lower $\delta^{15}N$ values. Increased water column oxygenation subjects sinking organic matter to more intense aerobic degradation, resulting in delivery of organic matter to the sediment that is lower in both amount and reactivity. Lower glacial rates of MSR and, consequently, more effective buffering of porewater sulfate concentrations and sulfur

isotopic compositions to those of seawater lead to less pronounced and less variable $^{34}S$-enrichment of porewater sulfate. This suggestion is supported by data from cores outside the modern OMZ (in deeper water), which display lower and less variable $\delta^{34}S_{pyr}$ values consistent with lower overall SRR[58], possibly reflecting less abundant and more refractory TOC.

According to the preceding discussion, the enhanced flux of reactive organic carbon to the SWI leads to a high rate of MSR across the Peruvian OMZ. Drawdown and isotopic distillation of porewater sulfate and accumulation of porewater sulfide result in $\delta^{34}S_{pyr}$ values between $\sim -30$ and $\sim -15$‰, which correlate with TOC concentrations (Fig. 4b) at the study site and in two additional cores within the OMZ (holes 680B and 686B). At the study site, $\delta^{34}S_{pyr}$ values correlate also with $\delta^{15}N$ values (Fig. 4a), as discussed above. However, some of the downcore samples, occurring mostly in interglacial layers, clearly deviate from this $\delta^{34}S_{pyr}$-TOC-$\delta^{15}N$ pattern (Figs. 2 and 4), with $\delta^{34}S_{pyr}$ values that are higher by ~10–20‰ than the trend at their corresponding TOC concentrations and $\delta^{15}N$ values. We suggest that these anomalously high $\delta^{34}S_{pyr}$ values reflect pyrite formation associated with the (paleo)SMTZ. The increase in $\delta^{34}S_{pyr}$ values does not appear to be accompanied by an increase in pyrite abundance (Supplementary Fig. 5d), approximated by the total chromium-reducible sulfur (TRS, see "Methods") yield. It is likely that our TRS extractions contain mineralogically and isotopically distinct sulfur fractions (i.e., acid-volatile, elemental and organic sulfur, and pyrite)[59]. We suggest that sulfide produced by AOM-SR in the SMTZ reacts with some of these sulfur fractions to produce pyrite that is isotopically distinct from the pyrite produced by the reaction of sulfide of OSR origin with iron-bearing phases. The sulfide produced by AOM-SR is expected to be more $^{34}S$-enriched because its sulfate source is already characterized by relatively high $\delta^{34}S$ values due to distillation during OSR and, as AOM-SR consumes sulfate quantitatively at the study site, it further distills porewater sulfate isotopes at a depth in the sediment where diffusive buffering of the concentration and isotopic composition of porewater sulfate is ineffective. Thus, reactions within the SMTZ could affect $\delta^{34}S_{pyr}$ values with little impact on the abundance of TRS, as observed here.

Multiple and distinct layers of barite and dolomite, which usually form at the SMTZ, have been reported at much shallower depths than the present SMTZ[27,34,35]. These shallow dolomite and barite layers have been interpreted as potential indicators of paleo-SMTZ positions[27,34,35]. If those paleo-SMTZs behaved similarly to the present one, we should expect $^{34}S$-enriched pyrite at depths of high Ba/Al ratio. However, except for the three samples located at

the present SMTZ, Ba/Al ratios do not appear to correlate with anomalously high $\delta^{34}S_{pyr}$ values (Supplementary Fig. 5k). We suggest that unlike $\delta^{34}S_{pyr}$, which may serve as robust a indicator of past locations of the SMTZ, reductive dissolution of barite may modify Ba/Al ratios[60,61], thereby obliterating records of the past SMTZ location.

A critical challenge in understanding Earth's surface evolution is differentiating between signals preserved in the sedimentary record that reflect global processes, such as the evolution of ocean chemistry, and those that are local, representing the depositional environment and the burial history of the sediments. These concerns are particularly relevant for records of sedimentary pyrite, which have been used to reconstruct global redox budgets and microbial metabolic evolution. Here, coupled C−N−S concentration and stable isotope data of glacial−interglacial sediments from the modern Peruvian margin reveal strong local environmental controls on sedimentary $\delta^{34}S_{pyr}$. Varying rates of microbial metabolic activity, regulated by regional oceanographic variations in OMZ extent and the flux of sinking organic matter, appear to have driven the observed $\delta^{34}S_{pyr}$ variability on the Peruvian margin. We note that this effect was unlikely due to large variations in cell-specific sulfate reduction rates, but due to variations in MSR rates at a community level, with an effect on porewater sulfate drawdown and isotopic distillation and on porewater sulfide accumulation and upward diffusion. Following the discovery and clear demonstration that local water-column oxygenation and organic matter deposition fluxes exert a strong control on $\delta^{34}S_{pyr}$, local variations in organic flux to the sediments may be listed alongside local variations in sedimentation rate[18,19,38,41] as drivers of stratigraphic variation in $\delta^{34}S_{pyr}$.

Increased organic carbon preservation, sedimentary lamination, and shifts in carbon isotope ratios (thought to reflect organic carbon burial), all associated with sedimentation under Peruvian OMZ conditions, have been widely documented in the sedimentary rock record. This suggests that the dynamics we describe here may be relevant to many geologic records of $\delta^{34}S_{pyr}$. In many cases, especially where $\delta^{34}S_{pyr}$ records are the sole constraint on sulfur cycling, similar stratigraphic variations in $\delta^{34}S_{pyr}$ have been used to infer changes to the marine sulfate reservoir driven by global-scale anoxia events[62–64]. In cases with $\delta^{34}S_{SO4}$ data accompanying $\delta^{34}S_{pyr}$, the offsets between the two measurements have been used to make inferences about the types/rates of microbial metabolisms that were active at the time (e.g., [65,66]). Such extrapolations rely on the assumption that changes in $\delta^{34}S_{pyr}$ data from individual locations reflect global-scale sulfur cycling, and the section studied here suggests there exists an important but previously overlooked control on $\delta^{34}S_{pyr}$ by the organic deposition flux.

This work adds to the growing body of evidence highlighting the local environmental controls that influence sedimentary pyrite $\delta^{34}S$ records. Such local controls need to be identified and evaluated before chemostratigraphic data can be used to reconstruct global biogeochemical cycling, the evolution of ocean chemistry, or Earth's surface redox state.

## Methods

**Pyrite content and sulfur isotopic composition ($\delta^{34}S_{pyr}$ values).** For pyrite sulfur, samples were extracted using the chromium reduction method[59,67]. This method allows the recovery of all reduced inorganic sulfur present in sedimentary samples (pyrite, element sulfur, and iron monosulfide phases). During extraction, samples were reacted with ~25 mL of 1 M reduced chromium chloride ($CrCl_2$) solution and 25 mL of 6 N HCl for 4 h in a specialized extraction line under a dinitrogen ($N_2$) atmosphere. The liberated hydrogen sulfide was reacted in silver nitrate (0.1 M) trap, recovering the sulfide as $Ag_2S$; reproducibility was under 5% for repeated analyses. Residual $Ag_2S$ was rinsed three times using Milli-Q water, centrifuged, and then dried thoroughly. Mass balance was used to calculate the amount of total reducible sulfur (TRS). The $Ag_2S$ powders were homogenized prior to analysis, when 450 μg was loaded into tin capsules with excess $V_2O_5$. The $^{34}S/^{32}S$

ratios of the $Ag_2S$ were measured on a Thermo Delta V Plus, following online combustion in a Costech ECS 4010 Elemental Analyzer, at Washington University in St. Louis (WUSTL). Pyrite sulfur isotope compositions are expressed in standard delta notation as permil (‰) deviations from the Vienna Canyon Diablo Troilite (VCDT), with an analytical error of <0.5‰ on replicate standards.

**Nitrogen and organic carbon content and isotopic composition ($\delta^{15}N$, $\delta^{13}C_{org}$ values).** Prior to organic carbon and nitrogen analyses, the carbonate fraction was removed from bulk samples by 48 h reaction with an excess of 1.0 N HCl following ref. [68]. During digestion, centrifuge tubes were placed in an ultrasonic bath to increase the mechanical separation of clay and calcium carbonates. Post-dissolution residues were washed three times with distilled water, centrifuged, and dried at 50 °C. The residual powders were homogenized, and prior to analyses, 50 mg were loaded into a tin capsule. Analyses were performed using an Elemental Analyzer (EA, Flash 2000—ThermoScientific) coupled to an isotope ratio mass spectrometer (Thermo Delta V Plus EA-IRMS) at WUSTL. Nitrogen isotope ratios are given in delta notation as permil deviations from air, whereas carbon isotope ratios are given in delta notation as permil deviations from the Pee Dee Belemnite (PDB), with an analytical error of <0.5‰ (2σ, replicate standards) for both nitrogen and organic carbon isotopes. TN and TOC concentrations were measured using the thermal conductivity detector of the ThermoScientific Flash 2000 at WUSTL.

## Data availability
All data generated in this study are provided in the supplementary materials, and have been deposited in the PANGAEA database.

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

## Acknowledgements

I. H. acknowledges support from a European Research Council Starting Grant No. 337183. V. P. is the recipient of a Dean of the Chemistry Faculty of the Weizmann Institute of Science Postdoctoral Fellowship. The authors acknowledge J. H., who kindly helped with sample preparation and data acquisition. S. C. is warmly thanked for discussions and for providing the XRF-dataset.

## Author contributions

V. P., D. A. F., and I. H. conceived the research. V. P. developed the age model and performed all chemical and isotopic analyses. All authors wrote the paper.

## Competing interests

Authors declare no competing interests.

**Additional information**

