## [Peer Review File · Nature Communications]

REVIEWER COMMENTS

Reviewer #1 (Remarks to the Author):

The study presents a dataset of 39 pyrite sulphur isotope data from Ocean Drilling Program Site 1229 on the Peru Margin in combination with total reduced sulphur content, total organic carbon and nitrogen, their isotopic compositions, as well as ratios of pyrite-bound iron over reactive and total iron, respectively. From the data it is concluded that higher organic carbon deposition due to the onset of oxygen minimum zones during interglacial times leads to less negative $\delta^{34}\text{S}$ values in pyrite.

I am glad to hear that the authors essentially concur with our previous interpretation (in Meister et al., 2019). Unfortunately, a great part of the presented material and discussion is not really novel. It represents to a large extent just a reproduction of our study, where TOC, $\delta^{34}\text{S}$, and reactive iron fractions were already analysed. In particular, the iron analytics has been done to much greater detail in our study, and some mechanisms were more thoroughly discussed compared to the presented manuscript. I feel that this work has not sufficiently been credited in the present manuscript.

Overall, I see that the study could become publishable after a revision, which should better address the findings of previous studies, in particular at Site 1229, and starting from there focus on the novel aspects.

Major comments

One novel aspect is that the XRD scan data from Contreras et al. (2013) were used to correlate the core with glacial-interglacial cycles from global $\delta^{18}\text{O}$ records. Even though this is beneficial, it is somewhat disturbing that the authors do not cite the reference in the main text (in line 107). Only by looking at the supplemental material (Fig. 1d) one finds out that the data are actually from the literature. The origin of the data should be clearly stated in the main text (not just the suppl. Material).

The data presented in the manuscript are at somewhat higher resolution (ca. meter-scale) which allows for a glacial-interglacial correlation. This is interesting as it allows for a distinction between glacial-interglacial scale variation and long-term variations seen in the longer $\delta^{34}\text{S}$ record of Meister et al. (2019). Modelling by Contreras et al. (2013) has shown that TOC variation is not sufficient to explain the glacial-interglacial cycle upward-downward variation of the sulphate-methane transition zone (and therefore probably also the high-resolution $\delta^{34}\text{S}$ variations shown here). A main factor is the sedimentation rate which changes rather dramatically at Site 1229. It is important to note that the water depth is very shallow and erosional gaps occur in glacial sealevel lowstand intervals. Only with strongly varying sedimentation rate it was possible to reproduce the high-amplitude variation of the SMTZ.

In contrast, higher $\delta^{34}\text{S}$ values below 30 mbsf may indeed be the result of higher TOC burial. However, the correlation of $\delta^{34}\text{S}$ with TOC in Fig. 4 is circumstantial. It is the result of organic matter still being degrading near the surface, which is misleading. The authors do not take into account that the TOC content below 30 mbsf was also higher in the past. The real trend between $\delta^{34}\text{S}$ and TOC can only be seen if a longer record is considered. Meister et al. (2019) provide measurements throughout the entire section, which reveals that high $\delta^{34}\text{S}$ values correlate with higher TOC content. This trend is the opposite of what the authors show in Fig. 4, and Fig. 4 is also counter-intuitive to their explanation in the text.

Minor comments

Title: "Diagenetic pyrite" (it is most likely not sedimentary, as the authors also discuss in the text)

Line 13: Rephrase entire sentence

Line 24: Rephrase “those”

Line 31: it is a “value” or “signature” (not really a fingerprint, which is a pattern)

Line 114: How did the authors account for erosional gaps?

Line 119: Please indicate the from which hole the samples were taken (1229A, B, C, D)? This is important because the different drill cores may show considerable offset.

Line 147: “microbial” or “diagenetic” controls

Line 149: which “local” depositional environments?

Line 154: “... there is no strong relationship between sedimentation rate and $\delta^{34}\text{S}$ (suppl Fig. 3).” I do not think it is possible to say that. The resolution of sedimentation rate at the individual cycle is not good enough. As mentioned above, there are erosional gaps in the core, as clearly demonstrated by Skilbeck and Fink (2006). These gaps clearly played a role, and they were instrumental in controlling the upward-downward migration of the sulphate methane interface. Furthermore, it has to be taken into account that the pyrite formation was likely offset with respect to the sediment surface, as the zone of reactive iron availability extends over a certain interval below the sediment surface.

Line 181: Reference to the origin of the data should be provided.

Line 191: The diagram in Fig. 3 is misleading. Even if some pyrite is derived from sulphide produced directly at the sediment surface, in contact with the bottom water, the isotopic composition represents a mixture. Especially during interglacial intervals, the SMTZ was very shallow (cf. Contreras et al., 2013), such that a fraction of the pyrite pool may have been influenced at least to some extent by sulphide produced in the sediment, perhaps even at the SMTZ.

Line 216: The sediment surface at Site 1229 was probably never below the OMZ (water depth only 150 m). During glacial lowstand the sediment surface was above the OMZ, with the latter being shifted further offshore (cf. Suess and von Huehne).

Line 218: I do not understand this: why should the rate of sulphide production exactly match the rate of consumption?

Line 219: Sulphide does obviously accumulate in the porewater of Site 1229 (D'Hondt et al., 2003).

Line 266: The aspect of fractionation was already discussed in detail in Meister et al. (2019), see references therein.

Reviewer #2 (Remarks to the Author):

The authors provide a new case study on the topic of local controls on pyrite sulfur isotopes, which has been proved by sedimentary records from the Gulf of Lion (PNAS) and New Zealand (SA) shelf. In general, the data

are reliable and the explanations are convincing, but there are several issues that need to be clarified before the manuscript is accepted for publication.

1) The authors state that OSR and AOM-SR are spatially separated (L43-45), however, to my knowledge, OSR could also occur in the SMTZ, see the paper (Jørgensen et al., 2019).

2) The authors state that high sedimentation rate leads to enrichment of 34-S (L149-152), which are the findings of their previous work, so references of 15 and 16 should be added. In addition, “noot” in the title of reference 16 should be “not”. Please check the spelling of all words carefully.

3) In L214-222, aqueous sulfide may become enriched in 34-S with progressive MSR, but their isotopic signals may not be recorded by formed pyrites when iron supply is limited.

Some of aqueous sulfide could be incorporated into organic matter or return to the seawater, and organic sulfur could also affect the pyrite sulfur isotope, please think about it.

4) During the glacial period, the authors suggest the porewater sulfate could be easily buffered by seawater due to low TOC content (L252-255), but their previous works suggest high sediment rate usually results in more “closed” system. Could the authors suggest a threshold value of sedimentation rate or TOC content to affect the communication between porewater and seawater sulfate??

5) Recent studies have shown that the nitrogen isotopes ($\delta^{15}\text{N}$) of organic carbon in SMTZ could be affected by anaerobic methanotrophic archaea (Hu et al., 2020). The data in this paper does not seem to have a similar pattern. Could the author further explain the relationship between the 34-S-enriched pyrite and nitrogen isotopes (L357-361)?

6) Ba/Al ratios could be influenced by many factors, thus it is not necessary that your sulfur data have to correspond to these ratios (L375-382).

In short, this is a very interesting paper, which shows that local factors can affect the sulfur isotope composition of pyrite through multiple processes, and more case studies are needed. I would like to recommend that this manuscript be revised and accepted for publication.

Best wishes,

Xiting Liu

References

Hu, Y., Feng, D., Peng, Y.B., Peckmann, J., Kasten, S., Wang, X.D., Liang, Q.Y., Wang, H.B. and Chen, D.F., 2020. A prominent isotopic fingerprint of nitrogen uptake by anaerobic methanotrophic archaea. *Chemical Geology*, 558.

Jørgensen, B.B., Beulig, F., Egger, M., Petro, C., Scholze, C. and Røy, H., 2019. Organoclastic sulfate reduction in the sulfate-methane transition of marine sediments. *Geochimica et Cosmochimica Acta*, 254: 231-245.

Reviewer #3 (Remarks to the Author):

Nature Communications manuscript NCOMMS-21-07636

Title: Sedimentary pyrite sulfur isotopes track the local extent and dynamics of the Peruvian Oxygen Minimum Zone

This manuscript by Pasquier et al. presents carbon-nitrogen-sulfur-iron geochemistry in the Ocean Drilling Program (ODP) Core 1229 offshore Peru. The authors find a positive relationship between total organic carbon content and sulfur isotopic composition of pyrite in the upper 40 m of sediments beneath an oxygen minimum zone (OMZ). The fluctuations of pyrite sulfur isotopic values are interpreted as a result of OMZ intensification/expansion and local enhancement of the organic deposition flux. The authors conclude that organic carbon loading plays a key role in determining pyrite sulfur isotopic values in ODP Core 1229.

The paper is well written, and the results add to the ever-growing database which clearly shows that in different depositional environments, the dominant control on pyrite sulfur isotopes can be very different (e.g.,

sedimentation rate, reactive iron availability, and organic carbon loading). This contribution highlights how localized conditions can affect the sulfur isotope signatures of pyrite, which has important implications for understanding the ancient sulfur isotope records. I believe this is suitable for Nature Communications.

I have only some minor suggestions below.

Kind regards,
Jiarui Liu

General comments:

Line 167-193: Figure 3a presents very interesting results of iron speciation. Since aqueous sulfide concentrations are probably close to the detection limit at the sediment–water interface (e.g., Fig. S6b), the bottom water of the Peruvian OMZ is most likely anoxic but not euxinic on a long-term basis. This contradicts with the previously proposed boundaries for distinguishing euxinic from ferruginous and oxic water columns as shown in Fig. 3a. However, I do notice that transient sulfidic events occurred in the water column (e.g., Scholz, 2018, Earth-Science Reviews). The Peruvian OMZ, therefore, shows so-called high-frequency temporal redox variation (Algeo and Li, 2020, GCA). It may be worthwhile to have a short discussion of the impacts of high-frequency redox variation on iron speciation.

Line 215, 219-222, and 232-236: I think the pooled $\delta^{34}\text{S}$ values of sulfides are lower than those of the instantaneously produced sulfide (cf. Mariotti et al., 1981, Plant and Soil). If this is correct, the logic here of having lower $\delta^{34}\text{S}$ values of sulfide in extra-OMZ sites does not work. I think the magnitude of their downcore increase depends on the extent of sulfate depletion (or depth-integrated net sulfate reduction rate). One may argue that in extra-OMZ sites, $\delta^{34}\text{S}$ values of porewater sulfide increase less steeply downcore due to a deeper SMTZ or no SMTZ. This does not at all invalidate the conclusions, but it can easily be misunderstood by someone who does not know the details.

Specific comments:

Line 16: “OMZ” should not be in the abbreviation form due to its first appearance.

Line 64, 214, and 256: I think it is somewhat misleading to say that sulfur isotopic evolution in marine sediments follows Rayleigh distillation. Strictly, Rayleigh distillation describes the irreversible reaction in a closed system. Marine sediments, however, are open to diffusion as stated on line 258 and below. Although the sulfur isotopic pattern of porewater sulfate and sulfide in marine sediments may be similar to that in a closed system, it's for different reasons. Consider rephrasing.

Line 188: In Liu et al. (2020, GCA), we used a 3:1 methanol:toluene mixture instead of pure methanol to more completely extract crystalline elemental sulfur. I think acid-volatile sulfur is more a proxy for iron monosulfides than polysulfides.

Figure 1: Panel c is too complicated. Separating the concentration data from the rate data would make it easier for the reader to understand the biogeochemical processes. If in revising this paper you choose to have one panel for concentrations and the other one for rates, can you please also add aqueous sulfide data on the plot? Moving the legend to the bottom may be helpful.

The location of dolomite layers in panels b and d is not consistent. Please check.

Supplementary Fig. 1: Labels a–d is inconsistent with figure caption.

Supplementary Fig. 2–3: “IODP leg 201” is a typo.

Supplementary Fig. 6: There are several errors in this figure. (1) Units of sulfide and sulfate are not nM. (2) It seems that the symbols of Sites 1229A and 1229D are incorrect. Core A has a much lower sampling resolution in the upper 10 m and some $[\Sigma\text{H}_2\text{S}]$ data of both cores seem to be missed (as plotted in ref. 8). (3) The reference of Site 1229A (labeled as \$, ref. 22; i.e., Mossmann et al., 1991) seems to be wrong. Please also check other published data.

The two x-axes of sulfide, sulfate, and methane have different ranges and should be explained in the figure caption (or in the figure legend). It would also be helpful if the SR rates and MG rates are shown in two separate panels.

Supplementary Table 1: Is the TS here the same as the TRS in the main text? I found the measurement of TRS or TS is not clear in the Methods section. Is the reduced sulfur content determined gravimetrically?

Reviewer #1 (Remarks to the Author):

The study presents a dataset of 39 pyrite sulphur isotope data from Ocean Drilling Program Site 1229 on the Peru Margin in combination with total reduced sulphur content, total organic carbon and nitrogen, their isotopic compositions, as well as ratios of pyrite-bound iron over reactive and total iron, respectively. From the data it is concluded that higher organic carbon deposition due to the onset of oxygen minimum zones during interglacial times leads to less negative $\delta^{34}\text{S}$ values in pyrite.

I am glad to hear that the authors essentially concur with our previous interpretation (in Meister et al., 2019). Unfortunately, a great part of the presented material and discussion is not really novel. It represents to a large extent just a reproduction of our study, where TOC, $\delta^{34}\text{S}$, and reactive iron fractions were already analysed. In particular, the iron analytics has been done to much greater detail in our study, and some mechanisms were more thoroughly discussed compared to the presented manuscript. I feel that this work has not sufficiently been credited in the present manuscript.

Overall, I see that the study could become publishable after a revision, which should better address the findings of previous studies, in particular at Site 1229, and starting from there focus on the novel aspects.

Response to Reviewer #1 summary: We are glad that the Reviewer evaluates our manuscript as publishable after revision. Though we indeed concur with much of the interpretation of Meister et al. (2019), we do note that the focus of these studies was different, and that our study adds several layers of novel information and insight, some of which are mentioned below.

Firstly, the development of an age model and sampling at a higher resolution permits examination of processes occurring over glacial-interglacial timescales, much shorter than the timescales resolvable by the lower-resolution data of Meister et al. (2019).

Secondly, Meister et al. (2019), did not report TOC concentrations from the same samples in which they measured $\delta^{34}\text{S}_{\text{pyr}}$ values. Instead, they based their $\delta^{34}\text{S}_{\text{pyr}}$ -TOC correlation on TOC measurements from several holes in site 1229 (please refer to Reviewer #1's comment 9). In contrast, for consistency with the age model and to allow direct comparison, we measured TOC concentrations in the same samples analyzed for $\delta^{34}\text{S}_{\text{pyr}}$ values. This allows the establishment of $\delta^{34}\text{S}_{\text{pyr}}$ -TOC correlations with much higher confidence, revealing the strong correlation and the anomalous behavior of the SMTZ samples (Fig. 4).

Thirdly, the coupling of nitrogen isotope and TOC data allows us to relate the processes controlling the isotopic composition of pyrite to glacial-interglacial variations in the extent and intensity of the OMZ. This aspect was not discussed at all in the study of Meister et al. (2019), in which the lower resolution of sampling did not reveal such variations.

Fourthly, though Meister et al. (2019) performed more detailed analysis of iron speciation, the level of iron speciation data obtained in our study allowed very similar inferences to be made about the formation of pyrite at the study site. Importantly, we compiled iron speciation data from other sites within and outside the OMZ and compared these data with our own iron speciation data. This has allowed generalization of our results to yield a better mechanistic understanding of the controls on pyrite sulfur isotopes not only at the study site, but in organic-rich depositional environments in general.

In summary, these and other marked differences make our study an important step forward in both mechanistic understanding of the controls on pyrite sulfur isotopes and generalization of the findings to other organic-rich environments.

Major comments

1. Comment: One novel aspect is that the XRD scan data from Contreras et al. (2013) were used to correlate the core with glacial-interglacial cycles from global $\delta^{18}\text{O}$ records. Even though this is beneficial, it is somewhat disturbing that the authors do not cite the reference in the main text (in line 107). Only by looking at the supplemental material (Fig. 1d) one finds out that the data are actually from the literature. The origin of the data should be clearly stated in the main text (not just the suppl. Material).

Response: We thank the reviewer for pointing out this referencing mistake, which we have corrected.

2. Comment: The data presented in the manuscript are at somewhat higher resolution (ca. meter-scale) which allows for a glacial-interglacial correlation. This is interesting as it allows for a distinction between glacial-interglacial scale variation and long-term variations seen in the longer $\delta^{34}\text{S}$ record of Meister et al. (2019). Modelling by Contreras et al. (2013) has shown that TOC variation is not sufficient to explain the glacial-interglacial cycle upward-downward variation of the sulphate-methane transition zone (and therefore probably also the high-resolution $\delta^{34}\text{S}$ variations shown here). A main factor is the sedimentation rate which changes rather dramatically at Site 1229. It is important to note that the water depth is very shallow and erosional gaps occur in glacial sealevel lowstand intervals. Only with strongly varying sedimentation rate it was possible to reproduce the high-amplitude variation of the SMTZ.

Response: We agree with the Reviewer that variations in sedimentation rate are expected to shift the position of the SMTZ. It is perhaps also correct (based on the modeling study of Contreras et al., 2013) that variations in TOC content are incapable of producing major shifts in the SMTZ on their own. We note, however, that large enough variations in TOC can shift the SMTZ considerably. For example, holding the sedimentation rate constant, a halving of TOC and an associated \sim halving of sulfate drawdown rates would lead to an approximate doubling of the sulfate penetration depth, and with it the SMTZ. We further note that the inferences made in the study of Contreras et al. (2013) were based on ^{14}C -derived sedimentation rates over the last few ka (sediments younger than late MIS 1; Skilbeck and Fink, 2006), whereas the variations observed and discussed here span the last 600 ka. Thus, the relevance of the ^{14}C -derived sedimentation rates to the interpretation of our results is unclear. The age model constructed in this study, agree well with the one developed by Schrader, 1992 (Supplementary Fig. 2), reveals only minor variations in sedimentation rate. We have added a figure to the Supplementary Information (Supplementary Fig. 5.j), and refer to this figure from the main text when discussing the potential effect of deposition rate on $\delta^{34}\text{S}_{\text{pyr}}$ variations (lines 126-127, 174).

Much of the above is beside the point, however, as there appears to be a misunderstanding of the proposed role of the SMTZ in explaining our observed $\delta^{34}\text{S}_{\text{pyr}}$ variations. We interpret the $\delta^{34}\text{S}_{\text{pyr}}$ variations to reflect the effect of varying TOC content

on the scale of sulfate drawdown and isotopic distillation, not the position of the SMTZ. We suggest that the SMTZ does contribute to late pyrite formation, and that this pyrite is ^{34}S -enriched. However, the $\delta^{34}\text{S}_{\text{pyr}}$ variations observed on glacial-interglacial timescales are, with the exception of 4 points (out of 39) that happen to coincide with the present-day SMTZ, unrelated to any evidence for the existence of a SMTZ, today or in the past (Fig. 2). Indeed, the 4 pyrite samples related to the present-day SMTZ fall off $\delta^{34}\text{S}_{\text{pyr}}\text{--TOC}$ and $\delta^{34}\text{S}_{\text{pyr}}\text{--}\delta^{15}\text{N}$ relationships, displaying $\delta^{34}\text{S}_{\text{pyr}}$ that are higher by ~10-20‰ than the trend at their corresponding TOC concentrations and $\delta^{15}\text{N}$ values (Fig. 4). **This suggests that the majority of the $\delta^{34}\text{S}_{\text{pyr}}\text{--TOC}$ covariation is unrelated to the SMTZ. So, while the SMTZ certainly generates ^{34}S -enriched pyrite, variations in its location are not the main driver of $\delta^{34}\text{S}_{\text{pyr}}$ variation at this site.**

Lastly, while variations in sedimentation rate in a diagenetic model may be necessary to achieve good agreement between modeled and measured sulfate and methane concentration profiles by shifting the SMTZ (Contreras et al., 2013), age model-derived sedimentation rates do not appear to correlate with the glacial-interglacial variations in $\delta^{34}\text{S}_{\text{pyr}}$ values that we observe (Supplementary Fig. 5j). We have clarified all of the above in the revised manuscript, including a more comprehensive comparison with the findings of Contreras et al. (2013) and Meister et al. (2019) (lines 113, 128-132, 205, 403-411).

- 3. Comment:** In contrast, higher $\delta^{34}\text{S}$ values below 30 mbsf may indeed be the result of higher TOC burial. However, the correlation of $\delta^{34}\text{S}$ with TOC in Fig. 4 is circumstantial. It is the result of organic matter still being degrading near the surface, which is misleading. The authors do not take into account that the TOC content below 30 mbsf was also higher in the past. The real trend between $\delta^{34}\text{S}$ and TOC can only be seen if a longer record is considered. Meister et al. (2019) provide measurements throughout the entire section, which reveals that high $\delta^{34}\text{S}$ values correlate with higher TOC content. This trend is the opposite of what the authors show in Fig. 4, and Fig. 4 is also counter-intuitive to their explanation in the text.

Response: We disagree with the Reviewer about the “misleading” nature of the $\delta^{34}\text{S}_{\text{pyr}}\text{--TOC}$ correlation presented in Fig. 4. The higher TOC concentrations in older sediments and the correlation with $\delta^{34}\text{S}_{\text{pyr}}$ values in the study of Meister et al. (2019) does not mean that organic matter availability does not drive shorter-timescale variations in $\delta^{34}\text{S}_{\text{pyr}}$ values, such as those we observe here.

We agree that organic matter is in all likelihood still being degraded near the surface, as well as at depth, albeit at a lower rate. This degradation of organic matter is the direct driver of the metabolic activities that are present in these and other sediments (sulfate reduction, methanogenesis) and the indirect driver of pyrite formation. At a given TOC concentration at the sediment-water interface, i.e., at a given rate of organic matter delivery to the sediments, the timescale for establishment of a steady state in the porewater sulfate, sulfide and methane concentration profiles is short, relative to the timescale of burial. This is especially so in sediments with such high TOC loads, in which reaction rates are high, leading to the establishment of a reactive-diffusive steady state on even shorter timescales.

At the above-mentioned steady state, most of pyrite formation is early, within the first few meters of sediment (e.g., Liu et al., 2021; Liu et al., 2020; Meister et al., 2019; this study). Continuous formation of pyrite deeper within the sediment, and in particular at the SMTZ is likely, as we clearly note in the discussion section of our manuscript subtitled ‘*Late pyrite formation in the sulfate-methane transition zone affects $\delta^{34}S_{pyr}$ values*’. However, most of the pyrite forms early and its isotopic composition here reflects the rapidity of sulfate drawdown and isotopic distillation near the sediment-water interface. As argued above, the rapidity of sulfate drawdown and ^{34}S enrichment is itself dependent on the TOC concentration. **Thus, the $\delta^{34}S_{pyr}$ –TOC correlation is not circumstantial, as suggested by the Reviewer, but indicative of the mechanism governing the glacial-interglacial $\delta^{34}S_{pyr}$ variation that we observe.**

We disagree also that Fig. 4 is counter-intuitive to our explanation. Higher TOC concentrations during deposition of the sediments (and, therefore, higher TOC concentrations even after degradation of some of the organic matter) should lead to more rapid drawdown and isotopic distillation of porewater sulfate, and to higher pooled product (pyrite) $\delta^{34}S$ values – exactly the pattern observed. To avoid confusion, we have clarified the relevant parts of the revised manuscript (lines 66-69, 233-249).

Minor comments

4. **Comment:** Title: “Diagenetic pyrite” (it is most likely not sedimentary, as the authors also discuss in the text)

Response: Pyrite formed in marine sediments during early diagenesis has been referred to as “sedimentary pyrite” for decades (e.g., Berner, 1970 “Sedimentary pyrite formation” and several hundred citing references). For consistency with other work coming out of the group, we prefer to keep the current terminology, but we clarify that by sedimentary we mean pyrite formed in sediments during early diagenesis (line 28), i.e., unlike syngenetic pyrite formed within the water column Li et al., 2011; Suits and Wilkin, 1998.

5. **Comment:** Line 13: Rephrase entire sentence.

Response: Done (lines 12-13).

6. **Comment:** Line 24: Rephrase “those”

Response: Done (line 24).

7. **Comment:** Line 31: it is a “value” or “signature” (not really a fingerprint, which is a pattern)

Response: We have replaced “fingerprint” with “signature” (line 32).

8. **Comment:** Line 114: How did the authors account for erosional gaps?

Response: According to Skilbeck and Fink (2001) two main erosional surfaces associated with coarser silt and fine sand are present in Hole A, 201-1229A-2H-6 and 201-1229A-6H-2 (here termed 2nd and 1st erosion surfaces, for convenience). After careful cross-correlation of the magnetic susceptibility in hole A, D and E

(Supplementary Fig. 3) it turns out that the 1st erosion surface is located below 45 m and is, therefore, not within the depth interval studied here.

Close visual examination of the 2nd erosion surface reveals the presence of a grey clay layer, but unlike in the deeper sections reported in Skilbeck and Fink (2001; e.g., 201-1229A-11H-6 or 201-1229A-14H-4), there is no evidence of shell debris or phosphatic hard ground, which are characteristic erosional features at the study site. Furthermore, according to our age model and the one developed by Schrader (1992) for Hole 681A (same drilling site), irrespective of whether one performs a depth cross-correlation, this section seems to be deposited during an interglacial interval. Hence, it is unlikely that an erosional feature affected the hole location during a period of high sea level. Instead, we propose that the greenish facies indicate a period of forced regression, without erosion at the study site, during the MIS7 lowstand.

In the revised SI, we describe our treatment of the erosional gaps (lines 28-40).

9. Comment: Line 119: Please indicate the from which hole the samples were taken (1229A, B, C, D)? This is important because the different drill cores may show considerable offset.

Response: For consistency with the developed age model, all samples used in this study come from Hole E. We agree with the Reviewer that some of the discrepancies between this study and the preexisting dataset of Meister et al. (2019) might be the result of drilling offsets (which were also not considered in Meister et al., 2019). To overcome this issue, in the revised manuscript and associated figures, a hole-to-hole correlation was accomplished by aligning the magnetic susceptibility measurements (Supplementary Fig. 3). All data available from site 1229 holes A and D can now be compared to hole E (i.e., mbsfE). The interpretations remain robust in this new correlation effort, which we describe fully in the text (lines 128-132).

10. Comment: Line 147: “microbial” or “diagenetic” controls.

Response: Done (lines 162-163).

11. Comment: Line 149: which “local” depositional environments?

Response: The local depositional parameters are mentioned in the following sentence (deposition rate), but we clarify further to avoid confusion (lines 164-168).

12. Comment: Line 154: “... there is no strong relationship between sedimentation rate and $\delta^{34}\text{S}$ (suppl Fig. 3).” I do not think it is possible to say that. The resolution of sedimentation rate at the individual cycle is not good enough. As mentioned above, there are erosional gaps in the core, as clearly demonstrated by Skilbeck and Fink (2006). These gaps clearly played a role, and they were instrumental in controlling the upward-downward migration of the sulphate methane interface. Furthermore, it has to be taken into account that the pyrite formation was likely offset with respect to the sediment surface, as the zone of reactive iron availability extends over a certain interval below the sediment surface.

Response: First, it is not necessary to capture the sedimentation rate within an individual cycle at a high resolution. If covariation between sedimentation rate and $\delta^{34}\text{S}_{\text{pyr}}$ values exists, and enough glacial-interglacial cycles are sampled (here 6-7), the covariation

should appear in a cross plot. The revised version of Supplementary Fig. 5j, now showing sedimentation rate on a logarithmic scale to reveal the full range of variation, demonstrates: (i) no systematic difference in sedimentation rate between glacial and interglacial intervals (both are characterized by sedimentation rates between ~ 0.01 and 0.1 m ka^{-1}), but a clear difference in $\delta^{34}\text{S}_{\text{pyr}}$ values; (ii) SMTZ samples are characterized by sedimentation rates on the high end, but similar to or even lower than some of the other samples, which do not display anomalously high $\delta^{34}\text{S}_{\text{pyr}}$ values. **This second point suggests mechanisms other than variations in sedimentation rate as the reason for the high SMTZ $\delta^{34}\text{S}_{\text{pyr}}$ values.**

Regarding the erosional surfaces and their effect on the $\delta^{34}\text{S}_{\text{pyr}}$ values measured at the study site, we refer the Reviewer to the response to comment 8 above. Briefly, two erosional surfaces have been suggested at the study site, one of which is outside the sediment depth range studied here, and one of which appears not to have been erosive at hole 1229.

Lastly, we agree with the Reviewer that pyrite formation is likely to continue to a certain depth below the sediment surface, and that this is likely to affect the isotopic composition of pyrite. In fact, we discuss these dynamics in the original manuscript, in the sections entitled '*Iron speciation suggests that most OMZ pyrite forms early*' and '*Late pyrite formation in the sulfate-methane transition zone affects $\delta^{34}\text{S}_{\text{pyr}}$ values*'. We have clarified this in the revised manuscript, where necessary.

13. Comment: Line 181: Reference to the origin of the data should be provided.

Response: Done.

14. Comment: Line 191: The diagram in Fig. 3 is misleading. Even if some pyrite is derived from sulphide produced directly at the sediment surface, in contact with the bottom water, the isotopic composition represents a mixture. Especially during interglacial intervals, the SMTZ was very shallow (cf. Contreras et al., 2013), such that a fraction of the pyrite pool may have been influenced at least to some extent by sulphide produced in the sediment, perhaps even at the SMTZ.

Response: We fail to see how Fig. 3 is "misleading". The figure simply shows the relationship between iron speciation, TOC concentrations and $\delta^{34}\text{S}_{\text{pyr}}$ values in samples from the sediment-water interface, from core tops (upper 0.5 meters of sediment), and from buried samples. On the basis of the data in this figure, it appears that a large fraction of the pyrite forms within the upper ~ 0.5 meters of sediment. The shallowest SMTZ in the model simulations of Contreras et al. (2013) occurs at a depth of ~ 3 meters below the seafloor (today the SMTZ is ~ 25 mbsf). Thus, even at its shallowest location, the effect of the SMTZ on core-top pyrite is expected to be indirect, through the effect of the SMTZ on the sulfate and sulfide concentration and associated isotopic composition profiles. This is the case for pyrite forming in Site 1229, where the sulfate concentration profile reflects rapid sulfate consumption at/near the SMTZ, and consequently, pooling and upward diffusion of sulfide. Indeed, the iron speciation-TOC- $\delta^{34}\text{S}_{\text{pyr}}$ figure (Fig. 3) to which the Reviewer refers, suggests that this is generally the case for intra-OMZ sites, but not for the extra-OMZ sites. This is one of the main points of the paper.

Importantly, all of the pyrite samples that are not at the present-day SMTZ form clear TOC– $\delta^{34}\text{S}_{\text{pyr}}$ and $\delta^{15}\text{N}$ – $\delta^{34}\text{S}_{\text{pyr}}$ arrays, and today's SMTZ pyrites fall off these arrays (see response to comment 3). **Thus, a direct effect of the SMTZ (and variations in its location) can be ruled out for most of the pyrite in the core.** We have clarified the above in the revised manuscript (lines 387, 407-411).

15. Comment: Line 216: The sediment surface at Site 1229 was probably never below the OMZ (water depth only 150 m). During glacial lowstand the sediment surface was above the OMZ, with the latter being shifted further offshore (cf. Suess and von Huehne).

Response: We agree with the Reviewer that Site 1229 was never below the OMZ, and this is exactly how we describe the OMZ dynamics in the original manuscript (lines 341-343). Regarding the arguments made in and around line 216 of the original manuscript, we note that whether the site was above or below the OMZ is immaterial, the important issue is whether the site is *within an OMZ* or not. The available surface-sediment Fe speciation data reveal fundamental differences between intra-OMZ and extra-OMZ sites. We suggest that these differences arise from the formation of pyrite from a “pooled” sulfide reservoir in intra-OMZ sites and from “instantaneous” sulfide in extra-OMZ samples. To avoid this confusion, in the revised manuscript we have clarified the distinction between intra-OMZ and extra-OMZ sites and emphasized that the location of extra-OMZ sites as being above/below the OMZ is unimportant (lines 193, 208 214-215, 220, 231, 241,249,259,265).

16. Comment: Line 218: I do not understand this: why should the rate of sulphide production exactly match the rate of consumption?

Response: The rate of sulfide production does not always match the rate of its consumption. In fact, we argue that when sulfide production outstrips its consumption, the sulfide accumulates in porewater, and that this may explain some of the higher $\delta^{34}\text{S}_{\text{pyr}}$ values that we observe in intra-OMZ sites (lines 217-221 in the original manuscript, lines 233-249 in the revised manuscript). However, in extra-OMZ sites, TOC in the zone of sulfate reduction is less abundant and less reactive. This results in lower OSR rates, and sulfide consumption (e.g., by reaction with iron) more easily matches the rate of sulfide production. In these environments, sulfide concentrations remain low and diffusive communication of sulfide between different depths in the sediment is minor. We have clarified this in the revised manuscript (lines 233-249).

17. Comment: Line 219: Sulphide does obviously accumulate in the porewater of Site 1229 (D'Hondt et al., 2003).

Response: We agree with the Reviewer (see the response to the previous comment). In Site 1229 and other intra-OMZ sites, sulfide clearly accumulates in porewater and we invoke this to explain some of the observed $\delta^{34}\text{S}_{\text{pyr}}$ values. In line 219 of the original manuscript, we are referring to sampling locations located outside of the present-day OMZ. We have clarified this in the revised manuscript (lines 195-198, 234-249).

18. Comment: Line 266: The aspect of fractionation was already discussed in detail in Meister et al. (2019), see references therein.

Response: Meister et al. (2019) do indeed surmise that no large fluctuations are expected in the microbial fractionation, which is likely close to 70‰ at Site 1229, as observed in many marine sediments. We now cite the study of Meister et al. (2019) in this context, and take several additional steps to support their supposition with data and discussion (in the original manuscript lines 266-308, and in the revised manuscript lines 291-333). The support is in the form of: (i) coupled community-level sulfate reduction rates and cell counts, allowing actual estimates of the cell-specific sulfate reduction rate, which is a major control on the microbial fractionation; and (ii) discussion of the influence of sulfide pooling on the apparent sulfate-pyrite offset, as opposed to the true sulfate-sulfide fractionation imparted by microbes. We conclude that section of the manuscript by noting that none of these lines of evidence is ironclad, and that in-situ isotopic analyses of pyrite are necessary to resolve the question of whether or not the microbial fractionation varied at the study site.

Reviewer #2 (Remarks to the Author):

The authors provide a new case study on the topic of local controls on pyrite sulfur isotopes, which has been proved by sedimentary records from the Gulf of Lion (PNAS) and New Zealand (SA) shelf. In general, the data are reliable and the explanations are convincing, but there are several issues that need to be clarified before the manuscript is accepted for publication.

Comments

19. Comment: 1)The authors state that OSR and AOM-SR are spatial separated (L43-45), however, to my knowledge, OSR could also occur in the SMTZ, see the paper (Jørgensen et al., 2019).

Response: We agree with the Reviewer. More accurately, metabolic zonation in the sediment should be pictured as a continuum between OSR and AOM-SR endmembers, in which these microbial pathways co-occur between regions that are dominated by one of the endmembers. We have clarified this in the revised manuscript (lines 45, 46-48).

20. Comment: 2)The authors state that high sedimentation rate leads to enrichment of 34-S (L149-152), which are the findings of their previous work, so references of 15 and 16 should be added. In addition, “noot” in the title of reference 16 should be “not”. Please check the spelling of all words carefully.

Response: Done.

21. Comment: 3) In L214-222, aqueous sulfide may become enriched in 34-S with progressive MSR, but their isotopic signals may not be recorded by formed pyrites when iron supply is limited. Some of aqueous sulfide could be incorporated into organic matter or return to the seawater, and organic sulfur could also affect the pyrite sulfur isotope, please think about it.

Response: We agree with the Reviewer, and the issue of iron availability and organic matter sulfurization was described and discussed in the original manuscript only a few lines below the lines mentioned in the Reviewer’s comment, as well as in other relevant places in the text (lines 250-264).

22. Comment: 4) During the glacial period, the authors suggest the porewater sulfate could be easily buffered by seawater due to low toc content (L252-255), but their previous works suggest high sediment rate usually results in more “closed” system. Could the authors suggest a threshold value of sedimentation rate or TOC content to affect the communication between porewater and seawater sulfate??

Response: The Reviewer is correct that, in previous work, we identified sedimentation rate (through its effect on diffusive versus advective delivery of sulfate) as an important control on $\delta^{34}\text{S}_{\text{pyr}}$ values. At Site 1229, however, the XRD- $\delta^{18}\text{O}$ age model developed in this study suggests no systematic glacial-interglacial differences in the sedimentation rate (Supplementary Fig. 5j). Thus, it appears that the variations in sedimentation rate at the study site were insufficient to drive major variations in $\delta^{34}\text{S}_{\text{pyr}}$ values. Indeed, the entire range of $\delta^{34}\text{S}_{\text{pyr}}$ values (-35 to -10% , excluding samples associated with the SMTZ) is observed at both the low and high ends of the sedimentation rate constrained by the age model. It is difficult without the development and application of a diagenetic model to constrain the exact threshold, and this threshold is expected to vary from location to location. Specifically at the study site, we think that sulfate drawdown and isotopic distillation due to the high TOC content occurs both at high and low sedimentation rates, and that sulfide accumulation and isotopic homogenization is an important control on $\delta^{34}\text{S}_{\text{pyr}}$ values. We clarify this in the revised manuscript (lines 164-176, 230-249).

23. Comment: 5) Recent studies have shown that the nitrogen isotopes ($\delta^{15}\text{N}$) of organic carbon in SMTZ could be affected by anaerobic methanotrophic archaea (Hu et al., 2020). The data in this paper does not seem to have a similar pattern. Could the author further explain the relationship between the 34-S-enriched pyrite and nitrogen isotopes (L357-361)?

Response: We thank the Reviewer for directing us to the study of Hu et al. (2020). We find it unlikely that, at their study site, the $\delta^{15}\text{N}_{\text{org}}$ values reflect archaeal biomass, which is expected to be negligible relative to the bulk organic matter (~ 1 w%) on which the N isotope analyses were made. Specifically, a cellular C content of ~ 380 fg C cell $^{-1}$ for an archaeal cell with a radius of ~ 1 μm (Khachikyan et al., 2019), and an upper limit on archaeal cell density in the sediment of 10^8 cells cm $^{-3}$, yield total archaeal C of 3.8×10^5 g C cm $^{-3}$, which is only $\sim 0.1\%$ of the TOC in the sediments studied by Hu et al. (2020). In our study site, where TOC concentrations are all greater than ~ 1 w%, the contribution of microbial biomass to the $\delta^{15}\text{N}_{\text{org}}$ values is expected to be even lower. We suggest that the observations of Hu et al. (2020) and our own observations almost certainly reflect water-column processes associated with N cycling. In the interest of remaining positive, we prefer not to counter the interpretations of Hu et al. (2020).

24. Comment: 6) Ba/Al ratios could be influence by many factors, thus it is not necessary that your sulfur data have to correspond to these ratios (L375-382).

Response: We agree with the Reviewer. Unlike $\delta^{34}\text{S}_{\text{pyr}}$ values, which may serve as robust indicators of past locations of the SMTZ, reductive dissolution of barite may modify Ba/Al ratios, thereby obliterating records of the past SMTZ location. We thank

the Reviewer for pointing this out and have added text to explain this in the revised manuscript (lines 408-411).

In short, this is a very interesting paper, which shows that local factors can affect the sulfur isotope composition of pyrite through multiple process, and more case studied are needed. I would like to recommend that this manuscript be revised and accepted for publication.

Best wishes,
Xiting Liu

Response: We thank Dr. Liu for this very helpful and positive review.

References

Hu, Y., Feng, D., Peng, Y.B., Peckmann, J., Kasten, S., Wang, X.D., Liang, Q.Y., Wang, H.B. and Chen, D.F., 2020. A prominent isotopic fingerprint of nitrogen uptake by anaerobic methanotrophic archaea. *Chemical Geology*, 558.

Jørgensen, B.B., Beulig, F., Egger, M., Petro, C., Scholze, C. and Røy, H., 2019. Organoclastic sulfate reduction in the sulfate-methane transition of marine sediments. *Geochimica et Cosmochimica Acta*, 254: 231-245.

Reviewer #3 (Remarks to the Author):

Nature Communications manuscript NCOMMS-21-07636

Title: Sedimentary pyrite sulfur isotopes track the local extent and dynamics of the Peruvian Oxygen Minimum Zone

This manuscript by Pasquier et al. presents carbon-nitrogen-sulfur-iron geochemistry in the Ocean Drilling Program (ODP) Core 1229 offshore Peru. The authors find a positive relationship between total organic carbon content and sulfur isotopic composition of pyrite in the upper 40 m of sediments beneath an oxygen minimum zone (OMZ). The fluctuations of pyrite sulfur isotopic values are interpreted as a result of OMZ intensification/expansion and local enhancement of the organic deposition flux. The authors conclude that organic carbon loading plays a key role in determining pyrite sulfur isotopic values in ODP Core 1229.

The paper is well written, and the results add to the ever-growing database which clearly shows that in different depositional environments, the dominant control on pyrite sulfur isotopes can be very different (e.g., sedimentation rate, reactive iron availability, and organic carbon loading). This contribution highlights how localized conditions can affect the sulfur isotope signatures of pyrite, which has important implications for understanding the ancient sulfur isotope records. I believe this is suitable for Nature Communications.

I have only some minor suggestions below.

Kind regards,
Jiarui Liu

Response: We thank Dr. Liu for the encouragement and for the helpful comments.

General comments

25. Comment: Line 167-193: Figure 3a presents very interesting results of iron speciation. Since aqueous sulfide concentrations are probably close to the detection limit at the sediment–water interface (e.g., Fig. S6b), the bottom water of the Peruvian OMZ is most likely anoxic but not euxinic on a long-term basis. This contradicts with the previously proposed boundaries for distinguishing euxinic from ferruginous and oxic water columns as shown in Fig. 3a. However, I do notice that transient sulfidic events occurred in the water column (e.g., Scholz, 2018, Earth-Science Reviews). The Peruvian OMZ, therefore, shows so-called high-frequency temporal redox variation (Algeo and Li, 2020, GCA). It may be worthwhile to have a short discussion of the impacts of high-frequency redox variation on iron speciation.

Response: We have added the requested information to the manuscript (lines 195-198 and 201-203).

26. Comment: Line 215, 219-222, and 232-236: I think the pooled $\delta^{34}\text{S}$ values of sulfides are lower than those of the instantaneously produced sulfide (cf. Mariotti et al., 1981, Plant and Soil). If this is correct, the logic here of having lower $\delta^{34}\text{S}$ values of sulfide in extra-OMZ sites does not work. I think the magnitude of their downcore increase depends on the extent of sulfate depletion (or depth-integrated net sulfate reduction rate). One may argue that in extra-OMZ sites, $\delta^{34}\text{S}$ values of porewater sulfide increase less steeply downcore due to a deeper SMTZ or no SMTZ. This does not at all invalidate the conclusions, but it can easily be misunderstood by someone who does not know the details.

Response: The Reviewer is correct, of course, that $\delta^{34}\text{S}$ values of pooled sulfide for a given fraction of sulfate remaining are lower than those of the instantaneously produced sulfide. However, in marine sediments in which sulfide accumulates (e.g., intra-OMZ sediments), sulfide from deeper within the sediment diffuses upwards and mixes with sulfide produced in-situ. The deep-sourced sulfide, which forms by reduction of more highly Rayleigh distilled sulfate, is more ^{34}S -enriched than the sulfide instantaneously produced in-situ. The isotopic composition of sulfide at any given depth then reflects the mixing of in-situ production and upward diffusion of more ^{34}S -enriched sulfide. In contrast, in extra-OMZ sediments, where little or no sulfide accumulates in porewater, we expect $\delta^{34}\text{S}_{\text{pyr}}$ values to have an isotopic composition bounded by those of the instantaneous and pooled sulfide expected at the fraction of sulfate remaining (Supplementary Fig. 8). We realize that this message did not come across clearly enough in the original manuscript, and have clarified this in the revised manuscript (lines 235-249).

Specific comments

27. Comment: Line 16: “OMZ” should not be in the abbreviation form due to its first appearance.

Response: Corrected, thanks.

28. Comment: Line 64, 214, and 256: I think it is somewhat misleading to say that sulfur isotopic evolution in marine sediments follows Rayleigh distillation. Strictly, Rayleigh

distillation describes the irreversible reaction in a closed system. Marine sediments, however, are open to diffusion as stated on line 258 and below. Although the sulfur isotopic pattern of porewater sulfate and sulfide in marine sediments may be similar to that in a closed system, it's for different reasons. Consider rephrasing.

Response: The Reviewer is correct, we have rephrased this in the reviewed manuscript.

29. Comment: Line 188: In Liu et al. (2020, GCA), we used a 3:1 methanol:toluene mixture instead of pure methanol to more completely extract crystalline elemental sulfur. I think acid-volatile sulfur is more a proxy for iron monosulfides than polysulfides.

Response: This has been corrected in the reviewed manuscript, thanks.

30. Comment: Figure 1: Panel c is too complicated. Separating the concentration data from the rate data would make it easier for the reader to understand the biogeochemical processes. If in revising this paper you choose to have one panel for concentrations and the other one for rates, can you please also add aqueous sulfide data on the plot? Moving the legend to the bottom may be helpful.

Response: We agree with the Reviewer and have split Figure 1 as proposed.

31. Comment: The location of dolomite layers in panels b and d is not consistent. Please check.

Response: The Reviewer is correct, we have corrected this.

32. Comment: Supplementary Fig. 1: Labels a–d is inconsistent with figure caption.

Response: The Reviewer is correct, we have corrected this.

33. Comment: Supplementary Fig. 2–3: “IODP leg 201” is a typo.

Response: We thank the reviewer for pointing out this labeling mistake, which we have corrected.

34. Comment: Supplementary Fig. 6: There are several errors in this figure. (1) Units of sulfide and sulfate are not nM. (2) It seems that the symbols of Sites 1229A and 1229D are incorrect. Core A has a much lower sampling resolution in the upper 10 m and some [ΣH₂S] data of both cores seem to be missed (as plotted in ref. 8). (3) The reference of Site 1229A (labeled as \$, ref. 22; i.e., Mossman et al., 1991) seems to be wrong. Please also check other published data.

Response: The Reviewer is correct, we have corrected all those issues.

35. Comment: The two x-axes of sulfide, sulfate, and methane have different ranges and should be explained in the figure caption (or in the figure legend). It would also be helpful if the SR rates and MG rates are shown in two separate panels.

Response: Done.

36. Comment: Supplementary Table 1: Is the TS here the same as the TRS in the main text? I found the measurement of TRS or TS is not clear in the Methods section. Is the reduced sulfur content determined gravimetrically?

Response: We now provide further TRS details in the Materials and Methods.

References

- Berner, R.A., 1970. Sedimentary pyrite formation. *American Journal of Science* 268, 1-23.
- Contreras, S., Meister, P., Liu, B., Prieto-Mollar, X., Hinrichs, K.U., Khalili, A., Ferdelman, T.G., Kuypers, M.M., Jørgensen, B.B., 2013. Cyclic 100-ka (glacial-interglacial) migration of subseafloor redox zonation on the Peruvian shelf. *Proc Natl Acad Sci U S A* 110, 18098-18103.
- Hu, Y., Feng, D., Peng, Y., Peckmann, J., Kasten, S., Wang, X., Liang, Q., Wang, H., Chen, D., 2020. A prominent isotopic fingerprint of nitrogen uptake by anaerobic methanotrophic archaea. *Chemical Geology* 558.
- Khachikyan, A., Milucka, J., Littman, S., Ahmerkamp, S., Meador, T., Burg, T., Kuypers, M.M., 2019. Direct Cell Mass Measurements Expand the Role of Small Microorganisms in Nature. *Applied and Environmental Microbiology* 85, 1-12.
- Li, X., Cutter, G.A., Thunell, R.C., Tappa, E., Gilhooly, W.P., Lyons, T.W., Astor, Y., Scranton, M.I., 2011. Particulate sulfur species in the water column of the Cariaco Basin. *Geochimica et Cosmochimica Acta* 75, 148-163.
- Liu, J., Antler, G., Pellerin, A., Izon, G., Dohrmann, I., Findlay, A.J., Røy, H., Ono, S., Turchyn, A.V., Kasten, S., Jørgensen, B.B., 2021. Isotopically “heavy” pyrite in marine sediments due to high sedimentation rates and non-steady-state deposition. *Geology*.
- Liu, J., Pellerin, A., Antler, G., Kasten, S., Findlay, A.J., Dohrmann, I., Røy, H., Turchyn, A.V., Jørgensen, B.B., 2020. Early diagenesis of iron and sulfur in Bornholm Basin sediments: The role of near-surface pyrite formation. *Geochimica et Cosmochimica Acta*.
- Meister, P., Brunner, B., Picard, A., Bottcher, M.E., Jørgensen, B.B., 2019. Sulphur and carbon isotopes as tracers of past sub-seafloor microbial activity. *Sci Rep* 9, 604.
- Schrader, H., 1992. Coastal upwelling and atmospheric CO₂ changes over the last 400,000 years: Peru. *Marine Geology* 107, 239-248.
- Skilbeck, C.G., Fink, D., 2006. Data report: Radiocarbon dating and sedimentation rates for Holocene–upper Pleistocene sediments, eastern equatorial Pacific and Peru continental margin. *Proceedings of the drilling program, Scientific Results* 201, 1-15.
- Suits, N.S., Wilkin, R.T., 1998. Pyrite formation in the water column and sediment of a meromictic lake. *Geology* 26, 1099-1102.

REVIEWERS' COMMENTS

Reviewer #1 (Remarks to the Author):

The authors have mostly addressed well my previous comments. I only see a few minor issues that could still be improved:

It should still be mentioned that the studied interval lies partially within the zone of proxy formation. The study of iron reactivity in Meister et al. (2019) has shown that iron pyritization can still proceed further in the uppermost tens of metres. Especially, slowly reacting clay-mineral bound Fe can still lead to further pyrite formation and potentially overprinting the measured high-resolution pattern. A possible way to test this could be by analysing an interval at greater depth at high resolution.

In line 175 it is written that the $\delta^{34}\text{S}_{\text{pyr}}$ values "are opposite in phase to those observed previously". Perhaps it should be explained better that this is not because the mechanism of $\delta^{34}\text{S}$ -signature formation is different, but because the depositional processes are different. Site 1229 is on the shelf at only 150 m waterdepth. During glacial lowstand, this site was certainly above the wavebase. The sedimentation rate is lower during glacial times (and may contain erosion surfaces), and accordingly, the TOC is also lower. Also comparing Fig. 1e and Fig. 2, it seems that sedimentation rates must have strongly changed.

Fig. 2. TOC should be plotted in Fig. 2, so that the covariance with $\delta^{34}\text{S}_{\text{pyr}}$ can be directly seen. Please also indicate "Site 1229" for Fig. 2.

References: check spelling of "Böttcher" and "Jørgensen"

Reviewer #2 (Remarks to the Author):

I have carefully read the revised manuscript, and according to their responses to my and other two reviewers' comments, I think this manuscript can be accepted and published. Xiting

Reviewer #3 (Remarks to the Author):

Pasquier et al. provide an improved manuscript, both in clarity and quality. All points raised by reviewers have been clearly addressed. I would only suggest one technical correction as below. After that, I am happy to see this manuscript published.

Line 212-212: Please change "a proxy for crystalline elemental sulfur" to "a proxy for elemental sulfur" or "a proxy for zero-valent sulfur". I am sorry that I was not clear in the last review, but the methanol/toluene mixture extracts both crystalline and amorphous elemental sulfur, as well as polysulfides. Pure methanol is usually enough, but toluene was added to make sure that most of the crystalline elemental sulfur was also recovered.

Best regards,
Jiarui Liu

Reviewer #1 (Remarks to the Author):

The authors have mostly addressed well my previous comments. I only see a few minor issues that could still be improved:

- 1. Comment:** It should still be mentioned that the studied interval lies partially within the zone of proxy formation. The study of iron reactivity in Meister et al. (2019) has shown that iron pyritization can still proceed further in the uppermost tens of metres. Especially, slowly reacting clay-mineral bound Fe can still lead to further pyrite formation and potentially overprinting the measured high-resolution pattern. A possible way to test this could be by analysing an interval at greater depth at high resolution.

Response: We thank the Reviewer for raising this point. To explore the possibility that relatively unreactive Fe contributed to the observations, we included the deeper samples (from 44 to 190 mbsf, n=17) from the study of Meister et al. (2019) in the Fe_{pyr}/Fe_{HR} statistics. We find that this does not change the mean Fe_{pyr}/Fe_{HR} value, and that it increases the standard deviation by only 0.01, suggesting negligible overprinting of our results by late pyrite formation. We clarified this in the revised manuscript (lines 210-211):

'Here, Fe_{pyr}/Fe_{HR} values stay essentially constant between depths of ~0.5 mbsf and the base of the borehole at ~190 mbsf (Supplementary Fig. 4), with an average value of 0.77 ± 0.06 .'

- 2. Comment:** In line 175 it is written that the $d34S_{pyr}$ values "are opposite in phase to those observed previously". Perhaps it should be explained better that this is not because the mechanism of $d34S$ -signature formation is different, but because the depositional processes are different. Site 1229 is on the shelf at only 150 m waterdepth. During glacial lowstand, this site was certainly above the wavebase. The sedimentation rate is lower during glacial times (and may contain erosion surfaces), and accordingly, the TOC is also lower. Also comparing Fig. 1e and Fig. 2, it seems that sedimentation rates must have strongly changed.

Response: We revised the manuscript to better explain this (lines 177-178):

'Moreover, despite an early diagenetic origin of the pyrite in both locations, the changes in $\delta^{34}S_{pyr}$ observed here (elevated and variable $\delta^{34}S_{pyr}$ during interglacial periods) are opposite in phase to those observed previously (elevated and variable $\delta^{34}S_{pyr}$ during glacial periods, associated with higher sedimentation rates).'

- 3. Comment:** Fig. 2. TOC should be plotted in Fig. 2, so that the covariance with $d34S_{pyr}$ can be directly seen. Please also indicate "Site 1229" for Fig. 2.

Response: We have modified the figure caption, as requested (line 754). However, for clarity of the graphics, we prefer not to show TOC in the same figure. Profiles of TOC and $\delta^{34}S_{pyr}$ are juxtaposed in Supplementary Fig. 4, where the covariance is clearly seen.

- 4. Comment:** References: check spelling of "Böttcher" and "Jørgensen".

Response: Done.

28. Meister P, Brunner B, Picard A, Böttcher ME, Jørgensen BB. Sulphur and carbon isotopes as tracers of past sub-seafloor microbial activity. *Sci Rep* 9, 604 (2019).

Reviewer #2 (Remarks to the Author):

I have carefully read the revised manuscript, and according to their responses to my and other two reviewers' comments, I think this manuscript can be accepted and published. Xiting.

Reviewer #3 (Remarks to the Author):

Pasquier et al. provide an improved manuscript, both in clarity and quality. All points raised by reviewers have been clearly addressed. I would only suggest one technical correction as below. After that, I am happy to see this manuscript published.

1. **Comment:** Line 212-212: Please change "a proxy for crystalline elemental sulfur" to "a proxy for elemental sulfur" or "a proxy for zero-valent sulfur". I am sorry that I was not clear in the last review, but the methanol/toluene mixture extracts both crystalline and amorphous elemental sulfur, as well as polysulfides. Pure methanol is usually enough, but toluene was added to make sure that most of the crystalline elemental sulfur was also recovered.

Best regards,

Jiarui Liu

Response: Done (line 216):

*'Iron speciation data and sulfur isotopic compositions of acid-volatile sulfur (a proxy for iron monosulfides), methanol-toluene extractable sulfur (a proxy for **zero-valent** sulfur) and pyrite similarly suggest early pyrite formation in the relatively organic-rich sediments in the Bornholm Basin⁴⁹.'*